# Alpha-glucans from bacterial necromass indicate an intra-population loop within the marine carbon cycle

Irena Beidler [1], Nicola Steinke[2,3], Tim Schulze[1], Chandni Sidhu[2], Daniel Bartosik [1,4], Marie-Katherin Zühlke[1], Laura Torres Martin[1], Joris Krull [1,2], Theresa Dutschei[5], Borja Ferrero-Bordera[6], Julia Rielicke[6], Vaikhari Kale[6], Thomas Sura[6], Anke Trautwein-Schult [6], Inga V. Kirstein[7], Karen H. Wiltshire[7], Hanno Teeling [2], Dörte Becher [6], Mia Maria Bengtsson [8], Jan-Hendrik Hehemann [2,3], Uwe. T. Bornscheuer [5], Rudolf I. Amann [2] & Thomas Schweder [1,4,7] ✉

Phytoplankton blooms provoke bacterioplankton blooms, from which bacterial biomass (necromass) is released via increased zooplankton grazing and viral lysis. While bacterial consumption of algal biomass during blooms is well-studied, little is known about the concurrent recycling of these substantial amounts of bacterial necromass. We demonstrate that bacterial biomass, such as bacterial alpha-glucan storage polysaccharides, generated from the consumption of algal organic matter, is reused and thus itself a major bacterial carbon source in vitro and during a diatom-dominated bloom. We highlight conserved enzymes and binding proteins of dominant bloom-responder clades that are presumably involved in the recycling of bacterial alpha-glucan by members of the bacterial community. We furthermore demonstrate that the corresponding protein machineries can be specifically induced by extracted alpha-glucan-rich bacterial polysaccharide extracts. This recycling of bacterial necromass likely constitutes a large-scale intra-population energy conservation mechanism that keeps substantial amounts of carbon in a dedicated part of the microbial loop.

Marine microalgae (phytoplankton) account for an estimated 40–50% of the global photosynthetic primary production[1]. Phytoplankton blooms in particular entail fixation of large amounts of carbon, considerable amounts of which are converted to various polysaccharides[2,3]. Secretion, leakage or lysis of algal cells release these polysaccharides as dissolved or particulate organic matter (DOM, POM), providing a diverse carbon and energy source for heterotrophic bacteria specialized in their uptake and degradation[4,5]. This utilization of dissolved algal glycans represents an essential part of the marine microbial loop[6] and thus the global carbon cycle. Particularly prominent in this process are marine members of the phylum *Bacteroidota*[7–9]. These bacteria target algal polysaccharides by specific

[1]Pharmaceutical Biotechnology, Institute of Pharmacy, University of Greifswald, 17489 Greifswald, Germany. [2]Max Planck Institute for Marine Microbiology, 28359 Bremen, Germany. [3]University of Bremen, Center for Marine Environmental Sciences, MARUM, 28359 Bremen, Germany. [4]Institute of Marine Bio-technology, 17489 Greifswald, Germany. [5]Biotechnology and Enzyme Catalysis, Institute of Biochemistry, University of Greifswald, 17489 Greifswald, Germany. [6]Microbial Proteomics, Institute of Microbiology, University of Greifswald, 17489 Greifswald, Germany. [7]Alfred Wegener Institute for Polar and Marine Research, Biologische Anstalt Helgoland, 27483 Helgoland, Germany. [8]Microbial Physiology and Molecular Biology, Institute of Micro-biology, University of Greifswald, 17489 Greifswald, Germany. ✉e-mail: schweder@uni-greifswald.de

sets of enzymes and transporters encoded in genomic islands, termed polysaccharide utilization loci (sg. polysaccharide utilization locus; abbr. PUL)[10]. PULs enable specialized *Bacteroidota* to thrive in close temporal succession to phytoplankton primary producers during algal blooms[11,12], providing a link to higher trophic levels as they are grazed on by, e.g., bacterivorous flagellates[13] and ciliates[14]. The high cell densities that are reached by some phytoplankton bloom-associated bacteria also render them susceptible to viral infections. It has been shown that phage numbers correspond well to bacterial cell counts during algal blooms, and that phages infect key polysaccharide degraders such as *Polaribacter* spp[15,16]. Viral lysis of such bacteria therefore fuels the organic matter pool by releasing dead bacterial organic matter, called necromass (i.e. derived from a living organism but no longer part of it), including its internal storage glucans[17]. However, it is so far unclear, whether or not under bloom conditions these released bacterial storage glucans are utilized simultaneously with dissolved algal polysaccharides.

Marine phototrophs use glucans as storage polysaccharides. Red algae (*Rhodophyta*), green algae (*Chlorophyta*) and dinoflagellates (*Dinophyceae*) form α−1,4 glucans with varying degrees of α−1,6 branching, while *Stamenopiles*, e.g., diatoms as well as haptophytes (*Haptophyta*), form β−1,3/ β−1,6 laminarin or chrysolaminarin[18]. It is reasonable to assume that marine heterotrophic bacteria utilize both storage glucan types. During previous studies of seasonal spring phytoplankton blooms off the North Sea island Helgoland in the southern German Bight, we found that most active and abundant planktonic members of the *Bacteroidota*, especially *Flavobacteriia* possess dedicated PULs for both, β-glucans and α-glucans[19]. In a study of 53 sequenced North Sea *Flavobacteriia* strains, we furthermore found laminarin and α-glucan specific PULs in 62% and 75% of the strains, respectively, whereas only 37% coded for alginate PULs[20]. Likewise, spring bloom metaproteome and transcriptome data obtained at Helgoland Roads revealed that glucan metabolism out-weighs that of any other polysaccharides in DOM[17,21]. However, while the high frequency and expression of laminarin-targeting genes in phytoplankton bloom-associated bacteria are readily explained by the abundance of algal laminarin, the role of α-glucans remains more elusive[5].

In a recent study we observed that bacteria associated with an overwhelmingly diatom-dominated spring algal bloom expressed their α-glucan PULs during peak bloom phases[17]. Since diatoms are not known to contain significant amounts of α-glucans, this led to the assumption that these bacteria may be specifically adapted towards recycling bacterial α-glucans of lysed bacteria. In this study, we show that dominant bloom-responding *Flavobacteriia* associated with phytoplankton blooms synthesize α-glucan storage polysaccharides while growing on algal biomass. At the same time, these bacteria likely take advantage of bacterial necromass such as released α-glucan from lysed bacterial community members, employing a specialized and conserved protein machinery for α-glucan utilization. This process likely acts as an intra-population energy conservation mechanism under bloom conditions that keeps a large amount of glucans in a loop and represents an as of yet neglected part of the marine carbon cycle.

## Results

### Sources of α-glucans during algal blooms
Recently, we conducted a study on the response of free-living planktonic bacteria (0.2–3 μm) to a diatom-dominated spring bloom at Helgoland Roads (southern German Bight, 54°11′N 7°54′E)[17]. Chlorophyll *a* and microscopic bacterial count data across a three-month time period from the beginning of March to the end of May 2020 revealed a close succession of two diatom-dominated bloom events, the first of which took place around the end of March until mid-April directly followed by a more pronounced second bloom event from the end of April until the end of May[17]. Major responders during these

blooms comprised members of the closely related flavobacterial clades *Polaribacter* and *Aurantivirga* (Fig. 1A, B).

Diatoms globally produce substantial amounts of β-glucans in the form of laminarin[5], but are not known to produce notable amounts of α-glucans[18]. To identify potential α-glucan sources, we analyzed 18S rRNA gene amplicon data obtained from >10 μm and 3–10 μm biomass size fractions from the 2020 spring bloom at Helgoland Roads (53 time points). Corresponding to microscopic biovolume data obtained in the framework of the Helgoland Roads time series, which has been gathering such data since 1975[17,22], 18S rRNA gene sequences confirmed the centric diatoms *Dytilum bightwellii* and *Ceratulina pelagica* as dominant microalgae during the first and second bloom events, respectively[17]. Importantly, algae with α-glucans such as *Rhodophyta*, *Chlorophyta* and *Cryptophyceae*[21] were rare (Fig. 1C). However, *Dinophyceae* (dinoflagellates), also known to contain α-glucans[18], were detected throughout the sampling period and continuously made up 7–47% of eukaryotic non-metazoan reads (>10 μm and 3–10 μm fractions). Dinoflagellate autotrophs (e.g. *Karenia* spp.) and heterotrophs (e.g. *Gyrodinium* spp.) were both abundant during the first bloom event (up to 27% and 20% of the eukaryotic non-metazoan reads, respectively), whereas heterotrophic species dominated among the dinoflagellates during the second bloom event (up to 30% non-metazoan reads), likely in response to higher bacterial numbers. As the flavobacteria only greatly responded during the second bloom event where the diatom *Ceratulina pelagica* dominated both 18S rRNA counts and previously collected biovolume data[17], we surmised that dinoflagellates are likely not a significant source of α-glucan to the bacteria. Likewise, choanoflagellates, also known to feed on bacteria, became more prominent during the main bloom phase (Fig. 1C, Supplementary Dataset 1). Grazing by heterotrophic flagellates therefore likely represents a factor that promotes the release of bacterial necromass, such as storage α-glucan, into the DOM pool.

### Marine Flavobacteriia degrade different types of α-glucans
In a previous study, we sequenced 53 coastal North Sea *Flavobacteriia* strains, 75% of which featured α-glucan PULs, more than for any other polysaccharide[20]. Analysis of PUL genes coding for carbohydrate-active enzymes (CAZymes) revealed a variety of CAZyme-rich genetic loci centered around one or more family 13 glycoside hydrolases (GH13) known to be involved in α-glucan degradation[23]. These loci also involved GH65, GH31 and GH97 genes previously found to be conserved in alpha-glucan PULs[20,24] alongside the characteristic *susC/D* gene pair encoding the recognition and uptake machinery (Supplementary Fig. S1).

Alignment of the corresponding SusD-like protein sequences together with α-glucan-PUL-associated SusD proteins obtained from bacterial MAGs of the 2020 spring bloom at Helgoland Roads[17] and the well-characterized α-glucan-binding SusD from *Bacteroides thetaiotaomicron* VPI-5482[25] revealed that the ligand-binding amino acids known from *B. thetaiotaomicron* were generally conserved. Conservation at these critical sites, especially in view of the overall low sequence identity (~30%), indicated that these proteins likely bind α-glucan (Fig. 2). Corresponding structure predictions revealed a separation into two distinct functional groups, one of which was characterized by an approximately 17 amino-acid-containing loop close to the binding site, similar to *B. thetaiotaomicron*[26] (Supplementary Fig. S2). The α-glucan PUL encoded SusD of *Flavimarina* sp. Hel_I_48 affiliated with this group. We heterologously expressed this SusD and could observe that it was held back by α-glucans such as pullulan and glycogen but not laminarin in native retarding gels, corroborating affinity of this group for α-glucans (Supplementary Fig. S3). The second group of SusD-proteins lacked the described loop along with two residues known to be ligand-binding in *B. thetaiotaomicron*. The result is a more open binding site that may serve as an adaptation to structurally distinct α-glucans (Supplementary Fig. S2). MAG-derived SusD sequences

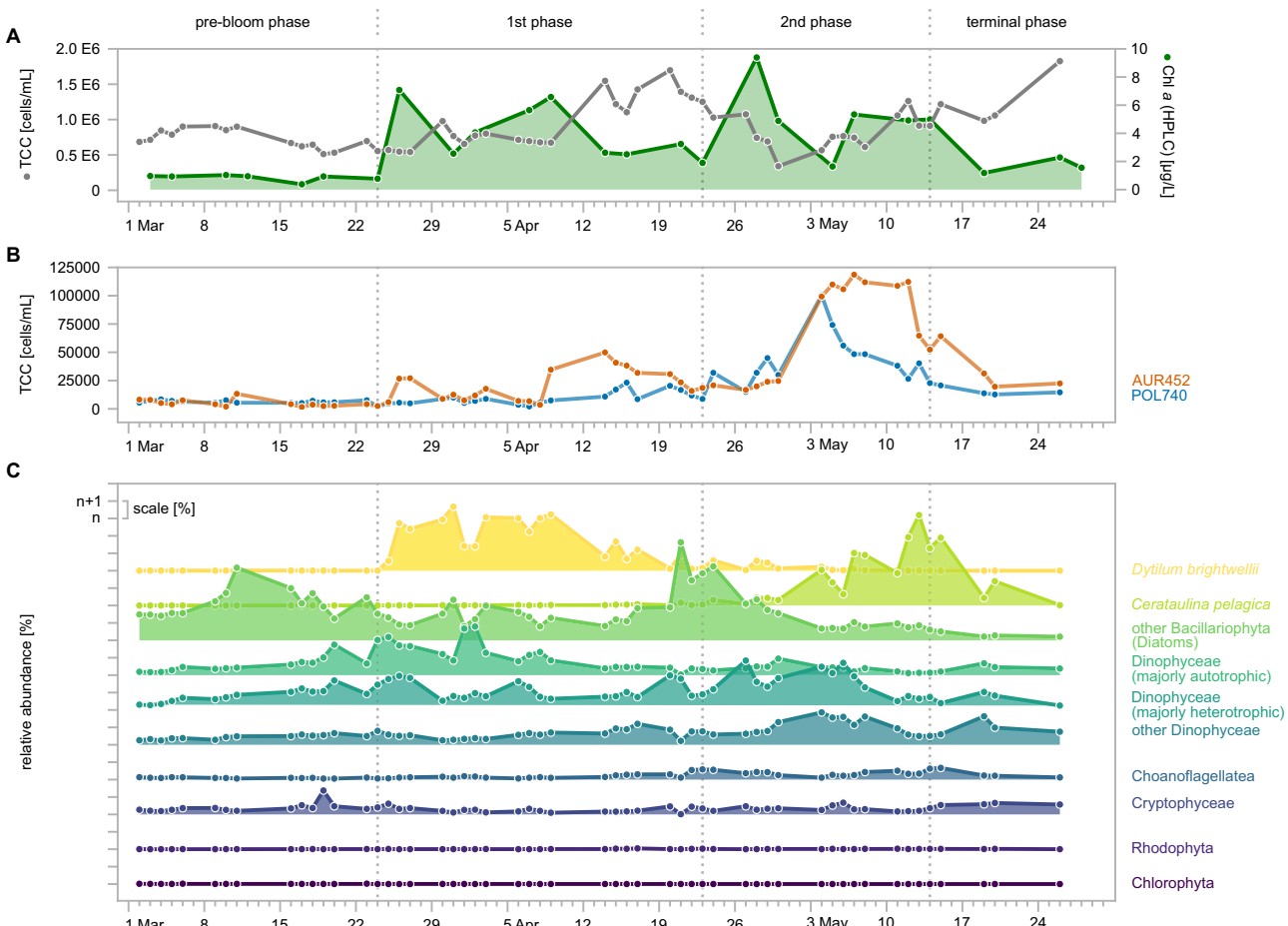

**Fig. 1 | Flavobacterial abundance and activity peaks align with increases in diatom abundance across the 2020 Helgoland spring phytoplankton bloom.** **A** Shown are total bacterial cell counts (TCC) and corresponding chlorophyll *a* data, which were taken from our published study by Sidhu et al.[17] to illustrate the dynamic of the investigated spring phytoplankton bloom. **B** Cell counts of the dominant flavobacterial clades *Polaribacter* (POL740) and *Aurantivirga* (AUR452) were detected by FISH. **C** Relative abundance of the main eukaryotic taxa as detected by 18S rRNA gene sequencing. Dominant diatom (*Bacillariophyceae*) taxa as well as largely heterotrophic and autotrophic *Dinophyceae* are plotted individually for clarity. Larger zooplankton was removed for this analysis. See also Supplementary Dataset 1. Figure was visualized using RawGraphs.

(0.2–3 µm fraction), including MAGs with detected expression during the 2020 bloom clustered with both groups, indicating ecological relevance of both PUL variants (Fig. 2). Gene composition analysis showed that PULs with an open type SusD almost exclusively coded for only few enzymes, e.g. only a sole GH13, whereas PULs with a looped SusD comprised a wider variety of CAZymes, most notably up to four GH13s as well as at least one SusE-like α-glucan-binding protein (Supplementary Fig. S1). The widest variety of GH13s was found in *Polaribacter* strains, representing one of the most recurrent bloom-associated bacterial clades at Helgoland Roads[12].

We conducted growth experiments with the North Sea strains *Polaribacter* sp. Hel_I_88 (α-glucan PUL with looped SusD) and *Muricauda* sp. MAR_2010_75 (α-glucan PUL with open SusD). These experiments revealed a correlation between α-glucan substrate complexity and growth efficiency when different α-glucans were offered as sole carbon source. While *Polaribacter* sp. grew well on glycogen and pullulan, which has a α−1,4- α−1,4- α−1,6 repeating unit, *Muricauda* sp. showed a preference for glycogen and grew poorly on pullulan (Supplementary Fig. S4). This demonstrates the presence of distinct α-glucans niches among marine bacteria.

**Marine bacteria contain multiple enzymes targeting α-glucans**
Protein sequence alignment of all PUL-encoded GH13s within 53 sequenced bloom-associated flavobacterial isolates showed that the *Polaribacter* sp. enzymes represented the majority, as roughly 70% of all isolate- and also MAG-associated GH13s grouped with them. Additionally, they represented three of the GH13s encoded in highly expressed bloom-associated MAGs (Fig. 3A, Supplementary Dataset 2). A notable exception was a GH13_31 with a proposed α−1,6 activity. While underrepresented in the isolates, the respective gene was highly expressed during the sampled 2020 bloom, indicating presence of this linkage type in marine bacterial α-glucans.

We heterologously expressed three GH13s encoded in the *Polaribacter* sp. Hel_I_88 PUL (Supplementary Fig. S1) in *Escherichia coli* and purified the enzymes for biochemical characterization. Via 3,5-dinitrosalicylic acid (DNS) reducing end assays, fluorophore-assisted carbohydrate electrophoresis (FACE) and thin layer chromatography (TLC), the enzymes GH13A (P161_RS0117435) and GH13C (P161_RS0117455) were shown to act on α−1,4 link-containing glycans glycogen and pullulan (Fig. 3B, Supplementary Fig. S5A–D). GH13A preferred glycogen, producing a dimer and smaller amounts of glucose, whereas GH13C was more active on pullulan, releasing primarily products with a degree of polymerization of three (dp3), likely panose or isopanose. The enzymes were inactive on α−1,6-linked dextran and β−1,3-linked laminarin. Selectivity for α−1,4 linkages was confirmed with malto-oligosaccharides. These oligosaccharides were degraded by both enzymes to dp2. In contrast, α−1,6-linked isomalto-oligosaccharides were not hydrolyzed. Experiments with mixed-link

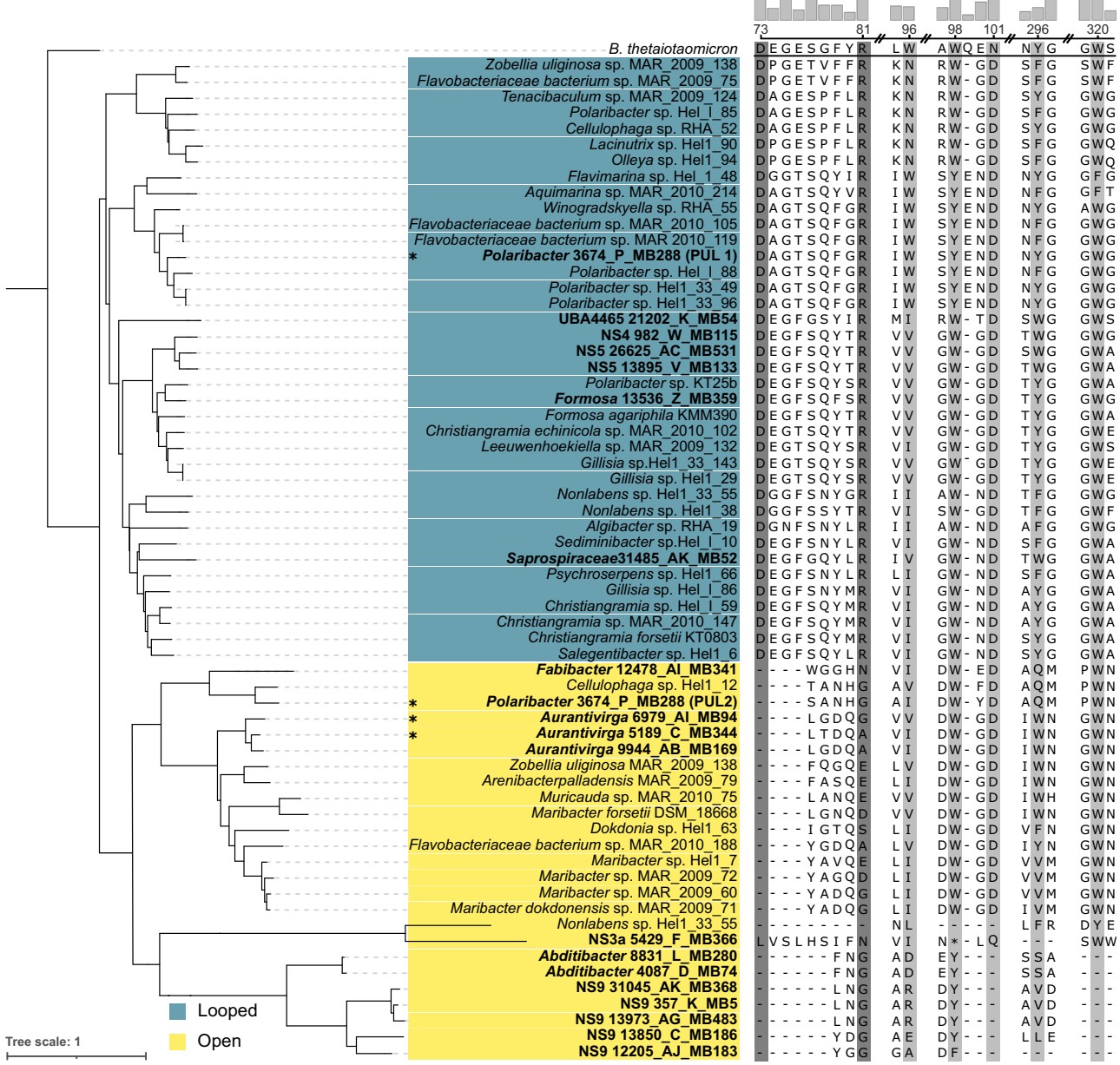

**Fig. 2 | The α-glucan-PUL encoded SusDs cluster in two groups defined by residues in binding positions.** Alignment of sequences from isolated North Sea *Flavobacteriia* and MAGs from the 2020 spring bloom at Helgoland Roads. Shown are residues in positions that were characterized as substrate binding in *B. thetaiotaomicron* SusD[25] (gray). Two groups, looped (blue) and open (yellow), were defined by two residues in binding position (dark gray) that are conserved in looped α-glucan SusD-like proteins but absent in open α-glucan SusD-like proteins. MAG sequences are highlighted in bold. An asterisk marks sequences that were found highly expressed in 2020 bloom metatranscriptomes. Conservation of amino acids at their position is represented as bars on top. Numbering corresponds to amino acids of *B. thetaiotaomicron* SusD.

α−1,6/α−1,4 oligosaccharides revealed minor activities on isopanose (α −1,4-α−1,6) and panose (α−1,6-α−1,4), indicating that an α−1,6-bond next to the α−1,4-connected glucose monomers at the −1 and +1 subsites hinders hydrolysis. Additionally, both enzymes acted on β-cyclodextrin (a ring of seven α−1,4-linked glucose units), releasing dp1 and dp2 (GH13A) and dp1, 2 and dp3 (GH13C), respectively. Interestingly, similar activity could not be detected on α-cyclodextrin (a ring of six α−1,4-linked glucose units), indicating that these enzymes recognize a specific substrate diameter (Fig. 3B, Supplementary Fig. S5A–D).

GH13B (P161_RS0117440) showed only minor activity on glycogen and pullulan, releasing a dimer (Fig. 3B, Supplementary Fig. S5B, D). Of the tested oligosaccharides with only one linkage type, only such containing α−1,4-linkages were acted upon, but all activities remained minimal. From isopanose, however, GH13B released notable amounts of dp1 and dp2, clearly indicating a preference for α−1,4-bonds situated next to α−1,6-bonds. No activity could be detected on panose, and as the enzyme released only dimers from polysaccharides, it can be assumed that the enzyme specifically releases isomaltose from the reducing end (Fig. 3B, Supplementary Fig. S5B, D). These results support an adaptation towards α−1,4/α−1,6-linked glucan substrates.

## Marine bacteria synthesize α−glucans

Marine bacteria are known to feature α-glucans as major storage polysaccharide. These α-glucans should therefore be formed during peak bloom phases, when excess organic carbon from algae outweighs the availability of other essential nutrients such as nitrogen,

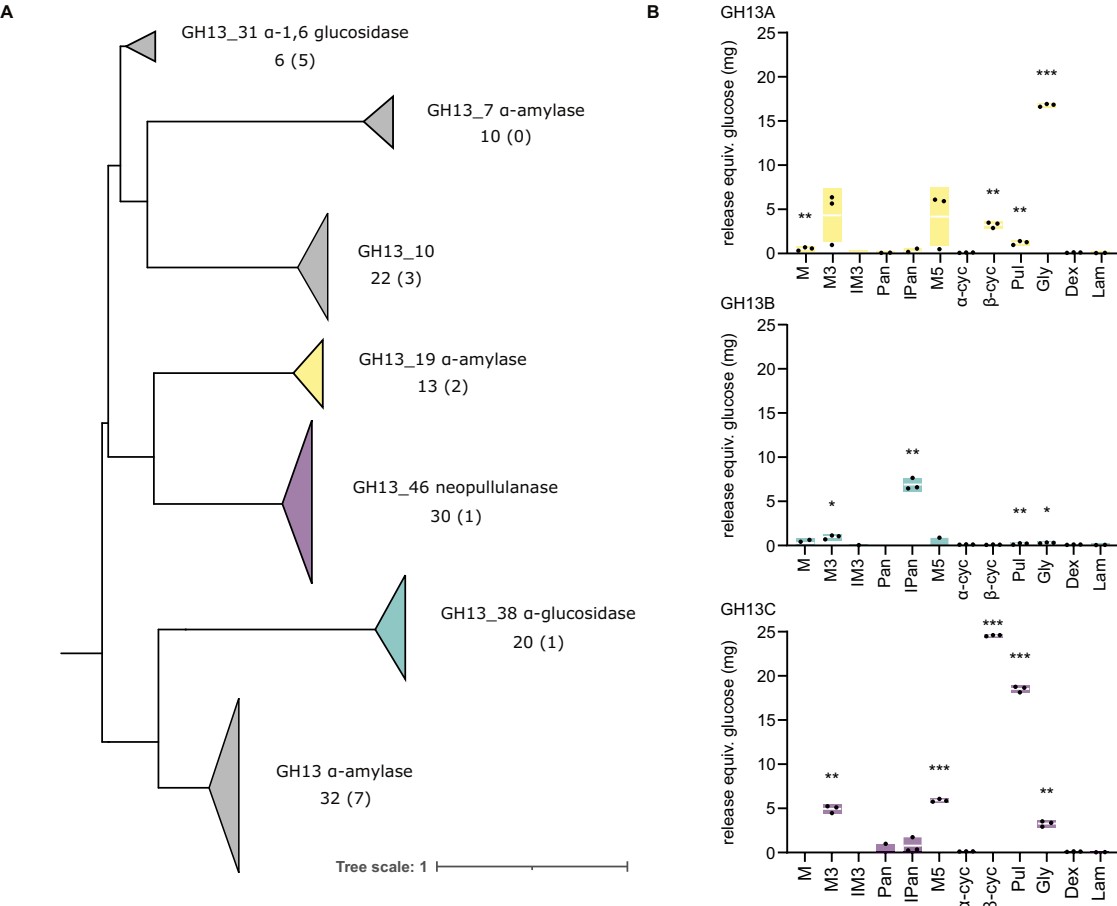

**Fig. 3 | Phytoplankton bloom associated bacteria encode a multitude of α-glucan-degrading enzymes. A** Maximum likelihood tree of the main GH13s encoded in the α-glucan PULs of 53 bloom-associated flavobacterial isolates as well as PUL-associated GH13s from the top 50 expressed MAGs of the 2020 Helgoland spring bloom[17]. Groups are clustered for clarity with arrows indicating group size. Numbers under the enzyme descriptions represent the number of sequences in the group with the included number of MAG-associated sequences in parenthesis. Colored arrows correspond to (**B**) characterization of representative *Polaribacter* sp. Hel_I_88 GH13 enzymes. 25 µg protein were incubated with 0.5% poly- or oligosaccharide for 24 h. Shown are all non-0 values (dots) with standard deviation

(bars) and mean (white lines) activity of recombinantly produced enzyme on different oligo- and polysaccharides measured via DNS-assay (all values corrected against oligo-/polysaccharide without enzyme, n = 3). Significance to controls containing laminarin with enzyme is indicated by asterisks above the respective bars (one-sided Student's *t* test, significant to *0.05, **0.005, ***0.0005). M Maltose, M3 Maltotriose, IM3 Isomaltotriose, Pan Panose, IPan Isopanose, M5 Maltopentose, α-cyc α-Cyclodextrin, β-cyc β-Cyclodextrin, Pul Pullulan, Gly Glycogen, Dex Dextrin, Lam Laminarin. See also Supplementary Fig. S5 for corresponding degradation patterns investigated via FACE and TLC.

as has been shown under nitrogen limitation in vitro[27]. Bacteria synthesize glucose-based storage polysaccharides via proteins encoded in the *glg*-operon, namely by addition of glucose-1-phosphate or maltose-1-phosphate to ADP-glucose (via GlgA or GlgE, respectively). This results in a linear α−1,4-glucan, to which α−1,6 linked branches are added by the branching enzyme GlgB[28].

Combined metagenome and metatranscriptome analysis of bacterial biomass from membrane filters with 0.2–3 µm pore size obtained during the spring bloom at Helgoland Roads in 2020 showed the dominant responder clades *Polaribacter* and *Aurantivirga* exhibiting high expression of their *glg*-genes that correlated with bloom progression. At the same time, both clades simultaneously expressed genes targeting laminarin and α-glucan (Fig. 4, Supplementary Dataset 3). This was corroborated by metaproteome analysis of bacterial biomass from 0.2 to 3 µm filters from spring blooms at Helgoland Roads in the years 2016, 2018 and 2020. These analyses reveal similar gene expression patterns that correlate with bloom progression as determined by chlorophyll *a* concentration measurements (Supplementary Fig. S6, Supplementary Dataset 3). Except for 2018, where the overall detection of laminarin-degradation proteins remained comparatively low (Supplementary Fig. S6A, B), a correlation of higher

abundances of laminarin-targeting proteins, Glg-proteins and α-glucan uptake proteins could be observed (Supplementary Fig. S6B, C). This suggests that α-glucan synthesis is a general mechanism of abundant bloom-associated *Flavobacteriia* in response to growth on algal laminarin-rich DOM. Consequently, when growing *Polaribacter* sp. Hel_I_88 on laminarin as sole carbon source, we detected a significant increase of α-glucan in these cultures over time via specific enzymatic hydrolysis of the polysaccharide extracts (Supplementary Fig. S7A). Thus, laminarin degradation and simultaneous α-glucan-synthesis could be confirmed with an isolated strain in vitro.

Proteomics revealed that proteins encoded by the *glg*-operon (P161_RS0109480, RS0109490, RS0109495 & RS0109500) were expressed continuously during growth on laminarin (Supplementary Fig. S8A). For instance, the abundance of GlgE maltose-1-phosphate maltosyltransferase increased significantly over the course of the experiment, indicating bacterial α-glucan formation.

As expected, overall protein abundance was dominated by the laminarin-PUL (P161_RS0117335-P161_RS0117415) (Fig. S8B). Yet, both the SusC/D-like protein pair (P161_RS0117480/85) and a GH13 (P161_RS0117500) of the α-glucan PUL became significantly more abundant during later growth phases (24 and 48 h), for which we could

also detect increased amounts of α-glucan in the bacterial culture (Supplementary Figs. S7A, S8C). A similar induction could not be shown for proteins of other PULs, such as the alginate PUL of *Polaribacter* sp. Hel_I_88 (P161_RS0107490- P161_RS0107540) (Supplementary Fig. S8D, Supplementary Dataset 3). These results support the view that, when bacteria lyse, the released necromass including bacterial α-glucan is sensed, taken up and utilized by the other bacteria. Analysis of 3–10 and 0.2–3 µm filter fractions sampled during the 2020 Helgoland spring bloom showed that α-glucans were more abundant on 0.2–3 µm filters, which largely represent the free-living bacterioplanktonic population. Concentrations rose to around 50 µg/L at the end of March, coinciding with the first bloom event. A spike of over 150 µg/L was observed at the beginning of May, coinciding with the main bloom event (Supplementary Fig. S9). Taken together, these results confirm that marine bacteria produce significant amounts of α-glucan storage polysaccharide during microalgal blooms and that the release of bacterial necromass upon lysis triggers recycling mechanisms including the induction of α-glucan-PULs in bloom-associated bacterioplankton species.

### Bacterial polysaccharide contains α-1,4-glucans

Monosaccharide analysis of enriched intracellular polysaccharide extracts from *Polaribacter* sp. Hel_I_88 cultures via high performance anion exchange chromatography with pulsed amperometric detection (HPAEC-PAD) revealed high proportions of glucose (29 mol%), while similar analysis of attached particles revealed higher amounts of glucosamine (Fig. 5A). Incubation of the enriched intracellular polysaccharide with recombinant *Polaribacter* sp. GH13A, GH13B and GH13C proteins showed visible degradation in reducing end assays. Combining all three enzymes, we observed a reducing end release of about 30% compared to extracts that were subjected to acid hydrolysis instead, corroborating that the extract contained considerable amounts of α-glucans (Fig. 5B). Correspondingly, a GH16 laminarinase from *Christiangramia forsetii* used as a control showed no activity, corroborating linkage-specificity of the extracted glucans. FACE-analysis of incubations with GH13A and GH13C yielded oligosaccharides of predominantly dp2, which was also the main product formed by incubation of either enzyme with glycogen. This was supported by GH13B releasing only dp2 from the extract, again indicating that the extracted polysaccharide indeed contained bacterial α-glucans (Fig. 5C).

### Bacterial polysaccharide induces α-glucan PUL expression

*Polaribacter* sp. Hel_I_88 grew on polysaccharide extracts from lysed cells as sole carbon source (Supplementary Fig. S4). Comparisons of culture lysate activity to glycogen or alginate-grown cultures showed a significantly increased activity on α−1,4-glucan-containing substrates for cultures that were grown on polysaccharide extract (Fig. 6A, Supplementary Fig. S10). Corresponding proteomics revealed a significant induction of nearly the entire α-glucan PUL (P161_RS0117430-P161_RS0117500) compared to alginate controls, with SusC and SusD proteins making up 1% and 0,46% of the entire proteome, respectively. Interestingly, this induction was higher than for a culture grown on glycogen as positive control, showing that the extracted polysaccharide elicited a more pronounced response (Fig. 6B, Supplementary Dataset 4).

While growth on extracted polysaccharide could also be observed for *Muricauda* sp. MAR_2010_75 (Fig. S4), a significant increase in extract-grown culture lysate activity could only be shown on glycogen (Fig. 6A). FACE analysis showed the degradation of α-glucan offered to the lysate under all tested conditions, indicating at least base-level activity of the α-glucan degradation machinery, regardless of growth conditions (Supplementary Fig. S10B). Proteomics revealed the α-glucan SusC- and SusD-like proteins (FG28_RS04375, FG28_RS04380) as most abundant during growth

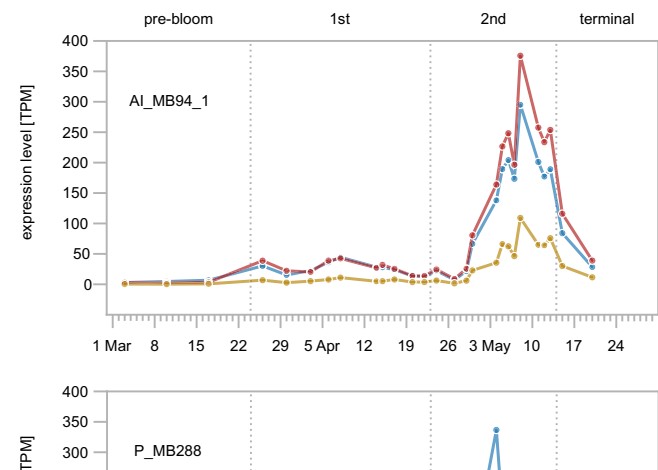

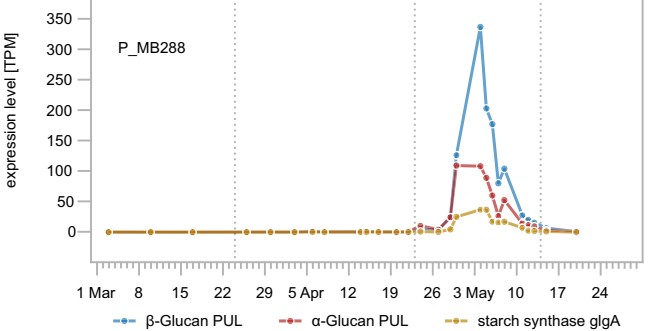

**Fig. 4 | Laminarin-degradation, α-glucan synthesis and α-glucan degradation occur simultaneously within dominant bloom responder clades.** Expression of the pathways was analyzed via transcripts mapped to the highest expressed MAGs of the clades *Aurantivirga* (AI_MB94_1) and *Polaribacter* (P_MB288). Shown are aggregated expression levels (Transcripts per million, TPM) of PUL-associated CAZymes of α-glucan and laminarin-targeting PUL-encoded genes compared to the expression of the starch synthase GlgA as proxy for the synthesis of α-glucan storage polysaccharides.

on the polysaccharide extract, but these were also highly expressed during growth on either glycogen or xylan. This high abundance was not mirrored by the PUL's associated CAZymes, a GH13 (FG28_RS04360) and a GH65 (FG28_RS04365), and suggests a difference in α-glucan utilization by bacteria with an open-type SusD-encoded α-glucan PUL (Fig. 6B, Supplementary Dataset 4).

## Discussion

The high cell densities of bacterioplankton during phytoplankton blooms entail increased mortalities as a consequence of elevated viral lysis and zooplankton predation[16,29]. The consequence is that bacteria are rather short-lived during bloom events. It has been estimated that during blooms about half of the bacterial biomass is recycled on a daily basis[30]. Hence, bacterial necromass is constantly released to the water column, including bacterial α-glucan storage polysaccharides.

As we show, both studied model bacteria can produce, take up and recycle these α-glucans. Prevalence and expression of corresponding Glg-proteins and α-glucan PULs in sampled bloom-associated *Bacteroidota* suggest that this is also a common and highly relevant process in situ which we could track to the dominant bloom responders *Polaribacter* and *Aurantivirga* via metatranscriptomics for the 2020 spring bloom at Helgoland Roads. It thus seems that during diatom-dominated phytoplankton blooms, bacteria employ a necromass recycling loop that is presumably constantly refueled by algal biomass such as β-glucans and bacterial α-glucans. While viral lysis and zooplankton predation diminish bacterial cell numbers, and predation furthermore shifts bacterial biomass towards higher trophic levels, rapid α-glucan recycling allows bacteria to achieve high growth rates during blooms and thereby to partially offset the loss due to increased mortality rates.

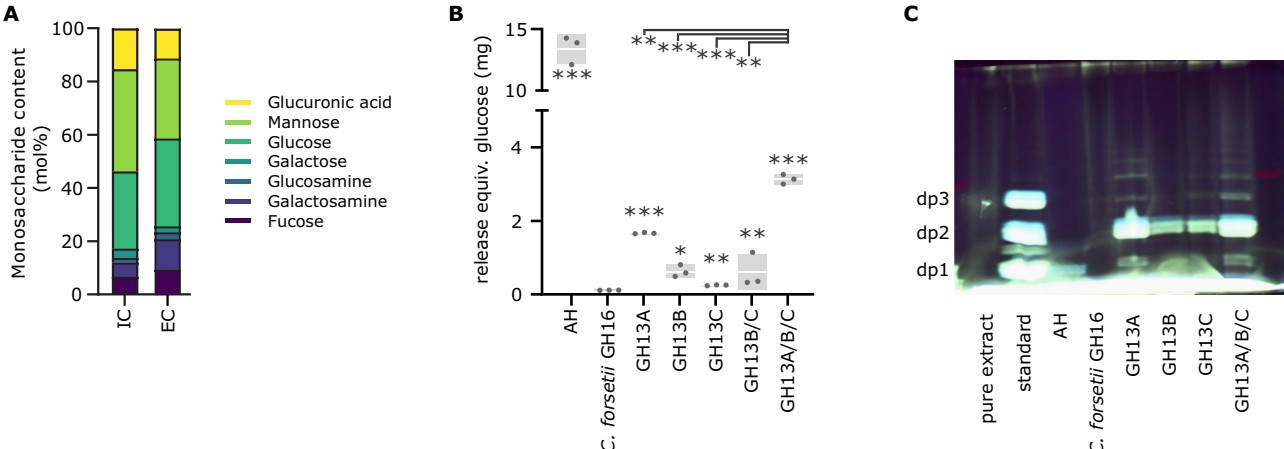

**Fig. 5 | Polysaccharide extracted from *Polaribacter* sp. Hel_I_88 contains high amounts of glucose in the form of α-glucan. A** Intracellular enriched extract (IC) or attached extracellular extract (EC) was subjected to acid hydrolysis (AH) and analyzed via HPAEC-PAD to determine monosaccharide composition (mol% of carbohydrate in the sample). IC was incubated with different recombinantly expressed *Polaribacter* sp. GH13s or *Christiangramia forsetii* GH16 laminarinase[88] and analyzed via (**B**) reducing end assay (25 µg of protein incubated with 0.5% extract solution for 24 h) or (**C**) FACE. Pure polysaccharide extract and extract after acid hydrolysis (AH) were used as controls. For FACE analysis, a mixture of glucose (dp1), maltose (dp2) and maltotriose (dp3) was used as standard. Values of the reducing end assay are corrected against untreated extract. All ($n = 3$) non-0 values are shown (dots) alongside the standard deviation (gray bar) and mean value (white line). Significance (one-sided Student's *t* test) is displayed as asterisks (significant to *0.05, **0.005, ***0.0005) either directly with the bar (to samples treated with *C. forsetii* GH16) or at the top (to a sample containing all tested *Polaribacter* sp. GH13s).

PUL analysis of North Sea *Bacteroidota* revealed two types of α-glucan PULs, a simpler PUL encoding less enzymes and a structurally open SusD (as in *Muricauda* sp. MAR_2010_75), which appears to target more simple, e.g. less branched or shorter α-glucans, and a more complex PUL encoding a looped SusD (as in *Polaribacter* sp. Hel_I_88 or *Flavimarina* sp. Hel_I_48), suggesting structurally more complex glucans as targets. The latter is more frequent in environmental metagenomes from bloom-associated North Sea bacteria and also regularly includes a SusE-like protein. SusE was shown to be starch-binding and essential in establishing the SusCD protein complex in *B. thetaiotaomicron* and may play a role in fine-tuning glycan uptake[31]. The two PUL types coded for diverse enzymes around one or more GH13 genes, supporting the view that the glucan substrate may exhibit structural variability. We showed activity on different predominantly α−1,4-linked glucans for representatives of three types of GH13s, all of which were present in prevalent MAGs obtained during phytoplankton bloom events. Since "looped"-SusD-type PULs commonly also code for GH97, GH31 and GH13_31 family enzymes, which have a proposed α−1,6-hydrolytic activity[32], this indicates an adaptation towards α−1,4/α−1,6-glucan.

Our results suggest that marine α-glucan cycling is unexpectedly complex and provides space for multiple distinct ecophysiological niches. Such niches may also include the decomposition of α-glucans that are produced by *Dinophyceae*[18,33]. We detected higher abundances of phototrophic *Dinophyceae*, such as *Karenia* spp., as well as higher amounts of α-glucans in bloom samples of 3–10 µm filters during the first bloom phase of the analyzed 2020 North Sea spring bloom. We suppose that marine bacteria are capable of also utilizing these microalgal α-glucans, as they are structurally similar to the α-glucans that are targeted by the enzymes we investigated[34]. However, close association of α-glucan PUL expression with peaks in diatom abundance led us to conclude that this was of minor importance during the analyzed diatom-dominated spring bloom.

The here described bacterial α-glucan loop is fueled by algal biomass. The simultaneous activity of α- and β-glucan PULs in pure culture experiments and during diatom-driven phytoplankton blooms in situ suggests that a large proportion of this turned-over biomass is laminarin. Laminarin is one of the most abundant macromolecules on earth[32]. It is not only produced from diatoms, but also from widespread *Prymnesiophyceae* such as *Phaeocystis* and coccolithophorids such as *Emiliania huxleyi*, which also form massive blooms[35]. Estimated production ranges around $12 \pm 8$ gigatons with a prevalence of $26 \pm 17\%$ in the particulate organic carbon pool[5]. Thus, intra-population bacterial α-glucan cycling is probably not a feature unique to diatom-dominated blooms. While other organic matter, such as proteins or other polysaccharides, can also be recycled, the synthesis of bacterial storage α-glucan from laminarin should be energetically favorable compared to its production from non-glucose glycans. In addition, laminarin-targeting PULs of marine *Bacteroidota* as for example in *Polaribacter* sp. Hel_I_88 and *Flavimarina* sp. Hel_I_48 often encode a GH149 β−1,3-glucan phosphorylase. This key enzyme has been shown to release glucose-1-phosphate from laminarin[36] which is the primary precursor of α-glucan synthesis.

Our data suggest that a substantial fraction of algal laminarin is not immediately remineralized, but rather converted to bacterial α-glucans. This means that α-glucans would likewise represent a significant portion of the marine carbon pool, also ranging in gigatons. Intra-population bacterial α-glucan cycling, in particular during phytoplankton blooms, therefore may constitute a substantial process within the global carbon cycle in terms of carbon turnover and fluxes that has so far not been well-recognized (Fig. 7). As such, the work presented in this manuscript represents an important building block that sheds light on bacterial intra-population carbon recycling. To verify the proposed model further studies should determine the flow of carbon from algal laminarin to bacterial α-glucan and its recycling, including the calculation of turnover rates from seawater measurements of different glucose polymers. This would also help to explore the specific niches occupied by bacteria with looped/open type SusD-containing PULs, further elucidating their role within this recycling loop.

The role of bacterial necromass turnover has been mainly demonstrated for terrestrial microbiomes, such as grassland soil ecosystems[37] or groundwater mesocosms[38]. Recently, the importance of such processes for the marine biogeochemical cycles could also be detected in marine sediments[39,40]. Here we show that such necromass turnover processes are likely also relevant during phytoplankton blooms and represent an important facet of the marine carbon cycle.

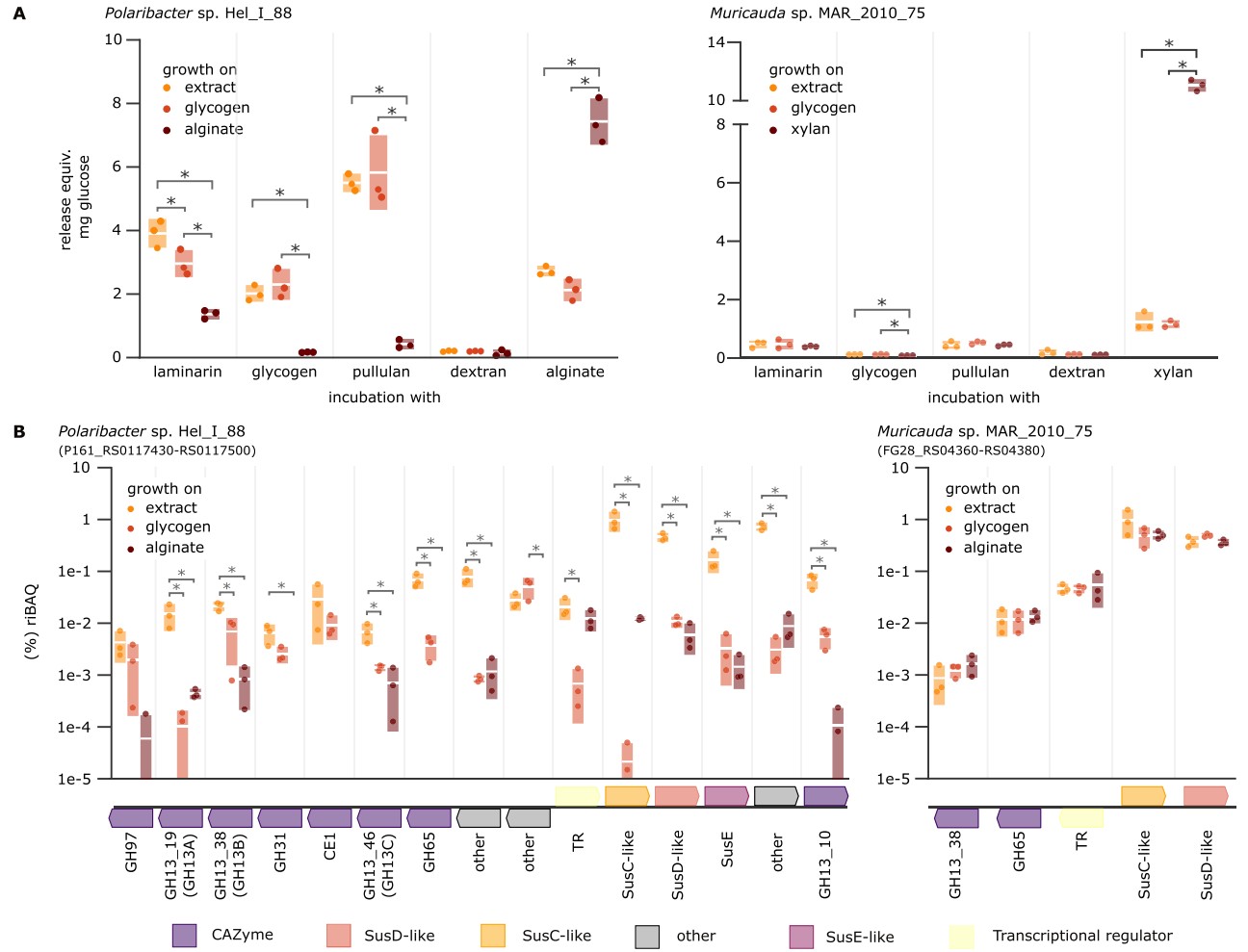

**Fig. 6 | The α-glucan PUL expression is specifically induced by bacterial α-glucan extracts. A** Lysate activity and α-glucan (**B**) PUL-encoded protein abundance of *Polaribacter* sp. Hel_I_88 and *Muricauda* sp. MAR_2010_75 grown on extracted bacterial polysaccharide (0.2% W/V) as sole carbon source. Samples were taken from biological replicates of growing cultures (*n* = 3) after 72 h and activity determined by incubating pure culture lysate (normalized for protein concentration determined via BSA-assay) with different polysaccharides. Alginate and xylan were used as controls, respectively. Shown are all non-0 values (dots) as riBAQ (relative identity-based absolute quantification) alongside their standard deviation (bars) and mean (white line). Significance determined by one-way ANOVA followed by post hoc Tukey's HSD test (*p*-value < 0.05) is displayed as asterisks above the respective values. The α-glucan PULs of both isolates are depicted beneath the protein abundances and specific CAZyme annotations are provided underneath.

Similar to the microalgal β-glucans, abundant bacterial storage α-glucans are likely highly soluble in water[41] and as such provide an accessible energy and carbon sources for planktonic bacteria with specific α-glucan-utilization machineries. Our findings suggest that uptake and recycling of bacterial α-glucans is a wide-spread intra-population energy conservation mechanism of abundant polysaccharide-degrading bacteria during phytoplankton bloom situations in the world's oceans.

## Methods

### Sampling site
Subsurface seawater (1 m depth) was collected at 52 time points between 2nd of March and 26th of May 2020 at the station Helgoland Roads near Helgoland in the southern North Sea. Since 1962 bucket water samples have been taken as part of a long-term monitoring program Helgoland Roads (54°11′N 7°54′E; DEIMS.iD: https://deims.org/1e96ef9b-0915-4661-849f-b3a72f5aa9b1)[42].

### Sequence analysis
Sequence analysis was carried out on the basis of the existing annotation[20] with additional reannotation as describes previously[21].

Sequences were aligned using ClustalO (v2.1)[43] in Unipro UGENE (v47)[44]. Trees were visualized using iTOL (v6.8.1)[45].

### Chlorophyll *a* measurements and cells counts of total bacteria and dominant bacterial clades
Sample filtration was carried out under dim light to avoid the pigment loss during the filtration procedure. For 2016 and 2018 samples, pigment extraction and analysis was carried out using a combined protocol from Zapata et al.[46] and Garrido et al.[4,47]. For 2020 samples, we followed the extraction and analysis method as described previously[48]. Subsequently, pigments were separated via high-performance liquid chromatography (HPLC) (Waters 2695 Separation Module), and detected with a Waters 996 Photodiode Array Detector (Waters, Milford, MA, USA). Total bacterial cell counts (TCC) and cell numbers of the dominant clades *Aurantivirga* (CARD-FISH probe AUR452) and *Polaribacter* (POL740) of the 2020 spring phytoplankton bloom were described and published previously[17].

### 18S rRNA gene sequencing
Sampled water was sequentially filtered onto polycarbonate membrane filters with different pore sizes (10 μm, 3 μm and 0.2 μm). For 18S

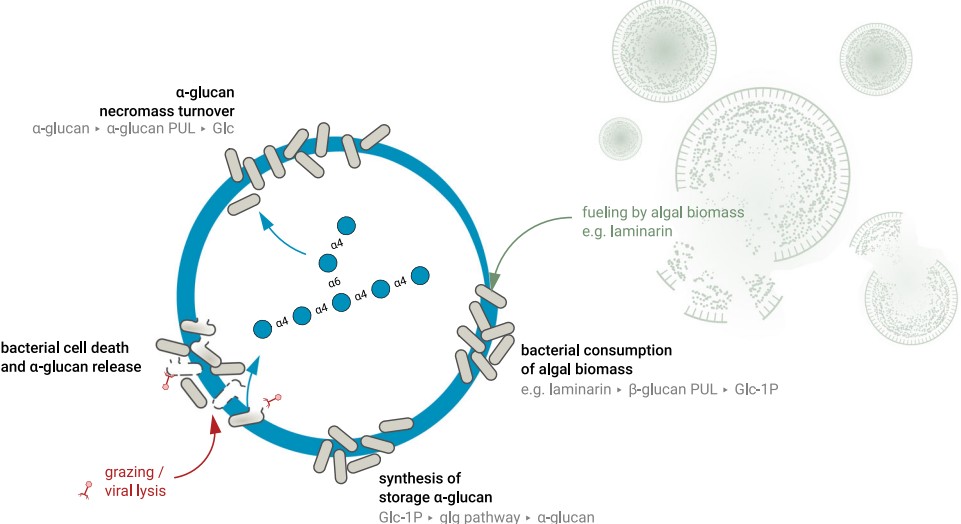

**Fig. 7 | Proposed succession model of the intra-population bacterial glucan flow during phytoplankton blooms.** Microalgae (such as diatoms) produce and release large amounts of biomass, including gigatons of laminarin. *Flavobacteriia* degrade this carbon source via their PUL-encoded enzymes, leading to the synthesis of bacterial storage α-glucan within the population. Viral infection and predator feeding cause lyses of a significant part of the bacterial population and thus releasing α-glucan storage polysaccharides as soluble DOM. This additional carbon source is recycled by the bacterial population using uptake and degradation pathways specifically adapted towards their own α-glucans. The α-glucan structure in the middle of the circle is shown in accordance with the symbol nomenclature for glycans[95].

rRNA gene amplicon sequencing, the two larger size fractions, >10 μm and 3–10 μm were analyzed. DNA was extracted from the filters using the DNeasy PowerSoil kit for DNA (Qiagen GmbH, Hilden, Germany). Mechanical lysis was achieved by bead beating in a FastPrep 24 5G (MP Biomedicals LLC, Irvine, CA, USA). The V7 region of the 18S rRNA gene was amplified using the primers [F-1183mod: 5′-AATTTGACT CAACRCGGG-3′, R-1443mod: 5′-GRGCATCACAGACCTG-3′][49] coupled to custom adapter-barcode constructs. PCR amplification and Illumina MiSeq library preparation (Illumina Inc., San Diego, CA, USA) and sequencing (V3 chemistry) was carried out by LGC Genomics in Berlin. Sequences have been submitted to the European Nucleotide Archive under the accession number PRJEB51816. Amplicon Sequence Variants (ASVs) were obtained using DADA2 (v1.26)[50] and taxonomically classified as described previously[51]. ASVs classified as *Metazoa* were removed before downstream analyses to reduce the effect of in particular crustacean zooplankton on community composition. For analysis, 10 μm and 3−10 μm counts were combined, set to 1 and relative abundances calculated. Classification of *Dinophyceae* into majorly heterotroph and majorly autotroph taxa was done for dominant groups according to literature[13,52−55].

## Metatranscriptomics

Metagenome and metatranscriptome sequencing were performed as described previously[17]. Briefly, thirty metagenomes were sequenced using PacBio Sequel II (Pacific Biosciences, Menlo Park, CA, USA) with one SMRT cell/sample while corresponding metatranscriptomes were obtained using Illumina HiSeq 3000 (~100 million reads/sample). The metagenomes were then processed to reconstruct metagenome-assembled genomes (MAGs), and mRNA reads were mapped to these genomes to identify highly expressed MAGs. Mapping and annotation were carried out using SqueezeMeta v1.3.1[56]. To predict open reading frames (ORFs), FragGeneScan v1.31 with the parameters "w1" and "sanger_5" as described by Rho et al.[57] was used. The predicted ORFs were searched against various databases including GenBank r239[58], eggNOG v5.0[59], KEGG r58.0[60], and CAZy (as of 11/12/2023)[61] using Diamond v0.9.24.125[62]. HMM homology searches against the Pfam 33.0 database[63] were conducted using HMMER3[64]. The combined

annotations were utilized for the manual prediction of polysaccharide utilization loci (PULs) and carbohydrate-active enzyme (CAZyme) clusters. Bowtie2[65] was employed to map mRNA reads to the MAGs, and transcripts per million (TPM) values were calculated for all MAGs in a given sample using the formula: (sum of reads successfully mapping to a MAG in the sample ×10^6)/(sum of contig lengths of the MAG × sum of reads in the sample).

## Metaproteomics

**Sample preparation.** The metaproteomics analysis of the 0.2 μm fraction from the spring phytoplankton bloom in 2016 has been described in detail in[21] and the free-living bacteria of the blooms from 2018 and 2020 were prepared as described previously[66]. Briefly, proteins were extracted from one-eighth of a filter (Millipore Express PLUS Membrane, polyethersulfone, hydrophilic, 0.2 μm pore size, diameter 142 mm) by cutting the filter into small pieces before transfer to 15 mL low binding tubes containing 1 mL resuspension buffer 1 (50 mM Tris-HCl (pH 7.5), 0.1 mg mL$^{-1}$ chloramphenicol, 1 mM phenylmethylsulfonyl fluoride (PMSF)) and 1,5 mL resuspension buffer 2 (20 mM Tris-HCl pH 7.5, 2% SDS (w/v)). After heating (10 min at 60 °C at 1000 rpm in a thermo-mixer), 5 mL DNAse buffer (20 mM Tris-HCl pH 7.5, 0.1 mg mL$^{-1}$ MgCl$_2$, 1 mM PMSF, 1 μg mL$^{-1}$ DNAse I) was added, and cells lysis was carried out by ultra-sonication (amplitude 51–60%; cycle 0.5; 3× 2 min) on ice before incubation for 10 min at 37 °C at 1000 rpm. After centrifugation (10 min at 4 °C at 10,000 × g), the supernatant was collected and the pelleted filter pieces were stirred and centrifuged again for 1 min at 4 °C at 5000 × g. Pre-cooled trichloroacetic acid (20% TCA (v/v)) was added for protein precipitation to the supernatant and after inverting the tube approximately 10x, the precipitate was pelleted via centrifugation (45 min, 4 °C, 12,000 × g) and the protein pellet was washed 3× in pre-cooled (−20 °C) acetone (10 min, 4 °C, 12,000 × g) before drying at room temperature. The proteins were resuspended in 2× SDS sample loading buffer (4% SDS (w/v), 20% glycerol (w/v), 100 mM Tris·HCl pH 6.8, bromphenol blue (tip of a spatula, to add color), 3.6% 2 mercaptoethanol (v/v) (freshly added before use)), incubated for 5 min at 95 °C before vortexing and separated via SDS-PAGE (Criterion TG 4–20% Precast Midi Gel, BIO-RAD Laboratories,

Inc., USA). The proteins were fixated, stained with Coomassie, and each gel lane was cut into 20 pieces[67]. Gel pieces were destained 3× for 10 min with 1 mL of gel washing buffer (200 mM ammonium bicarbonate in 30% acetonitrile (v/v)) at 37 °C under vigorous shaking, dehydrated in 1 mL 100% acetonitrile (v/v) for 20 min and the supernatant was removed before drying the gel pieces in a vacuum centrifuge at 30 °C. Proteins were in-gel reduced with 100 μL 10 mM dithiothreitol in 25 mM ammonium bicarbonate buffer (1 h at 56 °C) and alkylated with 100 μL 55 mM iodoacetamide in 25 mM ammonium bicarbonate buffer (without light for 45 min at room temperature) before the supernatant was removed. The gel pieces were washed with 1 mL 25 mM ammonium bicarbonate buffer (10 min, 1000 rpm at room temperature), dehydrated with 500 μL (2018 bloom) or 800 μL (2020 bloom) 100% acetonitrile for 10 min. The supernatant was removed before gel pieces were dried in a vacuum centrifuge (20 min) and finally covered with 120 μL trypsin solution (2 μg/mL Trypsin (Promega). After incubation for 20 min at room temperature, excess trypsin solution was removed and incubated in a thermo-mixer 15 h at 37 °C without shaking. Peptides were eluted with 120 μL solvent A (water MS grade in 0,1% acetic acid (v/v)) by sonication for 15 min before protein containing supernatant was transferred into a new tube. Peptide elution was repeated with 120 μL 30% acetonitrile (v/v) by sonication for 15 min. The eluates were pooled, and eluate volume was reduced in a vacuum centrifuge to a maximum of 15 to 20 μL. The peptides were desalted via ZipTips μC18 (Merck Millipore, P10 tip size) according to the manufacturer's protocol. The eluted samples were dried in a vacuum centrifuge and resuspended in 10 μL 0.5× Biognosys iRT standard kit in solvent A.

**LC-MS/MS measurement and data analysis.** Information about the LC MS/MS measurement and data analysis of the 0.2 μm fraction from the spring phytoplankton bloom in 2016 has been described in detail in ref. 21 and are described here for the free-living bacteria of the blooms from 2018 and 2020. An Easy-nLC1000 (Thermo Fisher Scientific, Waltham, MA, USA) was coupled to a Q Exactive mass spectrometer (Thermo Fisher Scientific) and peptides were loaded onto in-house packed capillary columns (20 cm length, 75 μm inner diameter) filled with Dr. Maisch ReproSil Pur 120 C18-AQ 1.9 μm (Dr. Maisch GmbH, Ammerbuch-Entringen, Germany) and separated using a 131 min non-linear binary gradient from 1% to 99% solvent B (99.9% acetonitrile(v/v), 0.1% acetic acid (v/v)) in solvent A (0.1% acetic acid (v/v)) at a constant flow rate of 300 nL min⁻¹. The MS1 scan was recorded with a mass window of 300–1650 m/z and a resolution of 140,000 at 200 m/z. The 15 most intense precursor ions were selected for HCD fragmentation (ions with an unassigned charge or a charge of 1, 7, 8, >8 were excluded) with a normalized collision energy of NCE 27. The resulting MS/MS spectra were recorded with a resolution of 17,500 at 200 m/z. Dynamic exclusion and lock mass correction were enabled.

All MS/MS spectra were analyzed using Mascot (version 2.7.0.1; Matrix Science, London, UK) and a bloom-specific metagenome-derived database containing all protein sequences from the 18 metagenomes obtained during the spring bloom in 2018 or 15 metagenomes obtained during the spring bloom 2020 assuming the digestion enzyme trypsin. Redundant proteins were removed using cd-hit[68] with a clustering threshold of 97% identity. The non-redundant database was added by a set of common laboratory contaminants and reverse entries, amounting to 81,874,922 (bloom 2018) or 4,221,978 (bloom 2020) sequences in the final database.

For database search with Mascot[69], the following parameters were used: fragment ion mass tolerance and parent ion tolerance of 10 ppm, none missed cleavages, variable modification on methionine (oxidation), and fixed modification on cysteine (carbamidomethylation). Scaffold (version 4.11.1 (bloom 2018) or version 5.0.1 (bloom 2020); Proteome Software Inc., Portland, OR) was used to merge the search results and to validate MS/MS-based peptide and protein

identifications[70]. During data analysis in Scaffold, an additional X! Tandem search was performed for validation (version 2017.2.1.4; The GPM, thegpm.org; version X!Tandem Alanine)[71] with default settings (fragment ion mass tolerance and parent ion tolerance of 10 ppm, carbamidomethyl on cysteine as fixed modification, Glu->pyro-Glu of the N-terminus, ammonia-loss of the N-terminus, Gln->pyro-Glu of the N-terminus and oxidation on methionine for 2018 and 2020 bloom, and additional carbamidomethyl of cysteine as variable modifications for 2018 bloom). Peptide identifications were accepted if they could be established at greater than 95% probability. Peptide probabilities from Mascot were assigned by the Peptide Prophet algorithm (bloom 2018)[72] or the Scaffold Local FDR algorithm (bloom 2020). Peptide Probabilities from X! Tandem were assigned by the Peptide Prophet algorithm[72] with Scaffold delta-mass correction. Protein identifications were accepted if they could be established at greater than 99% probability and contained at least two identified peptides. Protein probabilities were assigned by the Protein Prophet algorithm[73]. Proteins that contained similar peptides and could not be differentiated based on MS/MS analysis alone were grouped to satisfy the principles of parsimony.

For (semi-)quantitative analysis of 2016[21], 2018 and 2020 metaproteomic datasets, percent normalized weighted spectra (%NWS) were calculated by dividing Scaffold's 'Quantitative Value' for normalized, weighted (i.e. protein size-adjusted) spectra for each protein group, by the sum of all quantitative values for the sample. Average values were calculated from three biological replicates, including '0' for proteins that were not identified within a replicate. To make bacteria-specific %NWS readily comparable across all samples, all bacterial spectra were normalized to 100% (%BacNWS) using taxonomic assignment for protein groups provided by GhostKOALA v2.0[74] (genus_prokaryotes + family_eukaryotes + viruses database).

## Comparative genomics

MAGs from 2010-2012, 2016[21] (European Nucleotide Archive project accession PRJEB28156), 2018 (PRJEB38290) and 2020[17] (PRJEB52999) were dereplicated using dRep[75] v3.2.0 with minimum completeness of 70% and contamination lower than 5% at 0.95 ANI (average nucleotide identity). Protein sequences for representative MAGs were predicted with Prokka[76] v1.14.6. PULs, CAZymes, SusC-like and SusD-like proteins were predicted as described previously[21] using hmmscan v3.3.2 against dbCAN-HMMdb-V12 and diamond[62] v2.1.1.155 against CAZyDB.07262023 provided by dbCAN[77]. PULs were predicted with a sliding window of seven genes, i.e. the CAZymes and susC/D genes can only be seven genes apart for them to be considered part of the same PUL. GH13-encoding PULs were classified as „α-glucan-targeting", whereas PULs carrying combinations of GH3, GH16, GH17, GH30 and/or GH5 enzymes were annotated as „β-glucan-targeting". These substrate predictions were curated manually according to further PUL encoded CAZymes.

For identification of enzymes involved in α-glucan synthesis, K numbers were assigned to each sequence by GhostKOALA v2.0[74] (genus_prokaryotes + family_eukaryotes + viruses) and KofamScan[78] (ver. 01/04/2023, KEGG release 106) with an E-value ≤ 0.01. Proteins with hits for K00963, K00975, K00693, K00750, K16150, K16153, K13679, K20812, K00703, K16148, K16147, K00700 and K16149 were kept as part of the α-glucan synthesis pathway.

## Strain and cultivation conditions

We used the North Sea flavobacterial strains *Polaribacter* sp. Hel_I_88 (isolated from seawater off Helgoland island) and *Muricauda* sp. MAR_2010_75 (isolated from seawater at Sylt island), as model organisms[79]. For pre-cultures and polysaccharide extractions *Polaribacter* sp. Hel_I_88 and *Muricauda* sp. MAR_2010_75 were grown over night (20 °C, 200 rpm) in Marine Broth (MB 2216, Difco). *Polaribacter* sp. and *Muricauda* sp. were tested for growth on specific carbon

sources in MPM medium[80] containing 0.1% (w/v) of a single poly- or monosaccharide (Glycogen from Oyster: Merck; Pullulan: Thermo Fisher Scientific; Glucose: Roth) or 0.2% (w/v) bacterial polysaccharide extract. Growth was assessed via measurement of optical density at 600 nm. Cultures for proteome analysis were carried out in 25 mL MPM medium using biological triplicates.

## Proteomics of pure cultures

Cultures of *Polaribacter* sp. Hel_I_88 for time series sampling were grown in 100 mL batches. Cultures of *Polaribacter* sp. Hel_I_88 and *Muricauda* sp. MAR_2010_75 for extract characterization were grown in 25 mL batches. For time series sampling, 25 mL samples were sequentially filtered through 3 and 0.2 μm polycarbonate filters (Ø 47 mm, Merck) using a vacuum pump (PC 3002 VARIO, VACCU-BRAND) after 16, 24 and 48 h. Filters were stored at −80 °C until further use. Protein was extracted from ¼ of each 0.2 μm filter and prepared for mass spectrometry as described for metaproteomics but using 10% 1D-SDS polyacrylamide gels.

For extract characterization, cultures were grown on either polysaccharide extract, glycogen and alginate (Sigma-Aldrich) (*Polaribacter* sp.) or xylan from beechwood (Sigma-Aldrich) (*Muricauda* sp.). After 72 h, 25 mL cultures were harvested via centrifugation at 4000 × *g* and stored at −80 °C until further use. Protein was extracted by resuspending the pellet in 2 mL 50 mM TEAB buffer containing 4% (w/v) SDS. Samples were incubated at 95 °C and 600 rpm for 5 min, cooled on ice and sonicated (HD/UV 2070, Bandelin, Berlin, Germany) for 5 min. Debris was removed by centrifugation (14,000 × *g*, 10 min) and protein concentration was determined using the Pierce BCA Protein Assay Kit (ThermoFischer Scientific). Per sample, 25 μg protein were used. Proteins were separated on a 10% 1D-SDS polyacrylamide gel at 120 V for 90 min.

Samples were measured using an easy nLCII HPLC system applying a 100 min gradient coupled to an LTQ Orbitrap Velos mass spectrometer (Resolution 30,000, Scan range 300-1 700) (Thermo Fisher Scientific Inc., Waltham, MA, USA)[81]. Using MaxQuant[82], spectra were matched using a target-decoy protein sequence database with sequences and reverse sequences of *Polaribacter* sp. Hel_I_88 (NCBI ASM68793v1) or *Muricauda* sp. MAR_2010_75 (NCBI ASM74518v1) and common laboratory contaminants. A protein and peptide level FDR of 0.01 (1%) with at least two identified peptides per protein was applied. Only proteins that were detected in at least two replicates were classed as identified. Relative iBAQ (intensity based absolute quantification) values were manually calculated from automatically calculated iBAQs. Data and Results are available through the ProteomeXchange Consortium via the PRIDE partner repository (http://proteomecentral.proteomexchange.org)[83] with the identifier PXD043390. Statistical analysis for differential expression was performed in Perseus (2.0.1.1)[84] using one-way ANOVA followed by post-hoc Tukey's HSD test.

## Cloning, protein expression and purification

Genes coding for the proteins GH13A (P161_RS0117435), GH13B (P161_RS0117440) and GH13C (P161_RS0117455) of *Polaribacter* sp. Hel_I_88 (NCBI accession NZ_JHZZ01000001) were codon optimized and synthesized without their signal peptide by de novo gene synthesis (BioCat GmbH, Heidelberg, Germany). They were cloned into pET22b+ in *Escherichia coli* BL21 (DE3) for protein production. The *susD* gene (P162_RS13765) of *Flavimarina* sp. Hel_I_48 (NCBI accession: NZ_JPOL00000000) was amplified from genomic DNA via PCR (primers fwd: GTGTCTCGAGTTAATAACCTGGGTTTTGAGTCAGGTTT; rev: GAGAGGATCCTGAGAAT GATCTTGACGTAACCTTAGAG) and cloned into pET22b+ via restriction/ligation. Proteins were produced in 200 mL LB cultures (30 μg mL⁻¹ ampicillin) by induction with IPTG and incubation over night at 20 °C. Cells were harvested by centrifugation (5000 × *g*, 20 min), lysed using BugBuster Protein Extraction Reagent (Merck) (GH13s) or sonication on ice (3 × 2 min, 50%

cycle, SusD) and centrifuged (9500 × *g*, 20 min) to remove debris. Proteins were purified by loading the lysate onto a prepacked 5 mL IMAC column (HisTrap HP 5 mL, Cytiva) equilibrated with IMAC Buffer A (100 mM NaCl, 20 mM Imidazole, 20 mM Tris-HCl, pH 8) using an ÄKTA Pure 25 L (Cytiva). Proteins were eluted with a step gradient of IMAC Buffer B (100 mM NaCl, 500 mM Imidazole, 20 mM Tris-HCL, pH 8). *Flavimarina* sp. SusD was further purified using size-exclusion chromatography (Superdex 200 Increase 10/300 GL, Cytiva) using SEC-Buffer (100 mM NaCl, 20 mM Tris-HCL, pH 7.4). The GH13 A, B & C proteins were desalted using a prepacked sepharose-based desalting column (HiPrep Desalting 26/10, Cytiva) with PBS Buffer (pH 7.4). All proteins were concentrated via spin columns (Pierce Protein Concentrator PES, 30K MWCO, 2–6 mL, Thermo Fischer).

## Enzyme characterization

Activity profiles for all enzymes were generated by 3,5-dinitrosalicylic acid (DNS) reducing-end assay[85] as well as fluorophore-assisted carbohydrate electrophoresis (FACE)[8]. 25 μg purified protein were incubated with 0.5% (w/v) poly-/oligosaccharide (dextran 70, maltose, maltotriose, isomaltotriose, and maltopentose from Roth; laminarin from *Laminaria digitata*, α-/β-cyclodextrin from Sigma; panose and isopanose from Megazyme) for 24 h. Samples were heat-inactivated at 80 °C for 10 min and centrifuged (13,000 × *g*, 10 min) to remove precipitated protein.

For the reducing-end assay, samples were incubated with DNS-reagent solution (30% (w/v) Potassium sodium tartrate tetrahydrate, 10 mg mL⁻¹ DNS, 0.4 M NaCl) for 15 min at 95 °C and cooled to RT before measurement at 540 nm (Infinite 200 PRO M PLEX, Tecan, Männedorf, Switzerland). Values were compared against those of solutions containing only polysaccharide or only enzyme. Statistics of reducing-end assays were performed using a one-way Student's *t* test with an FDR of ≥0.05.

FACE was performed with 8-aminonaphthtalene-1,3,6-trisulfonic acid (ANTS) as fluorophore. 100 μL of the reaction samples were dried in a SpeedVac (Concentrator Plus, Eppendorf) and dissolved in 4 μL 0.05 M ANTS (in DMSO, 15% (v/v) acetic acid) and 4 μL 1 M NaCNBH₃ (in DMSO). They were incubated over night at 37 °C before being loaded onto a FACE-Gel[86] and separated at 400 V for 1 h.

**Thin-layer chromatography.** 25 μg purified protein were incubated with 0.5% (w/v) poly-/oligosaccharide in PBS for 24 h. Samples were heat-inactivated at 80 °C for 10 min and centrifuged (13,000 × *g*, 10 min) to remove precipitated protein. Glucose, maltose and maltotriose (all 1 mg mL⁻¹) in PBS were used as standard for the chromatography. The samples were analyzed as described previously[87] on silica gel plates (60 F245) with a mixture of 1-butanol, acetic acid and water (2:1:1) as solvent. Plates were sprayed with staining solution (4 g α-diphenylamine, 4 mL aniline, 200 mL acetone, 30 mL phosphoric acid 80% (v/v)) and visualized by heating above 100 °C.

**Affinity gel electrophoresis.** Gel electrophoresis was carried out using native 12% acrylamide-gels as described previously[88]. Purified *Flavimarina* sp. SusD was diluted with 10 mM Tris-HCL pH 7.4 to decrease the NaCl concentration to 20 mM. *Flavimarina* sp. SusD and BSA, which was used as non-binding control, were loaded onto gels containing either no additive or 0.2% of the tested polysaccharide. Runs were performed at 80 V with cooled buffer on ice. Gels were stained using Coomassie Brilliant Blue.

**Protein structure prediction.** Structure models of *Flavimarina* sp. Hel_I_48 (Supplementary Dataset 5) and *Muricauda* sp. MAR_2010_75 SusD (Supplementary Dataset 6) were predicted using ColabFold[89] using default settings with the top-ranked structure relaxed (num_relaxed: 1). These models were used for an overlay with the structure of the experimentally characterized homolog from *Bacteroides*

*thetaiotaomicron* in complex with cyclodextrin to determine differences between the SusD binding sites. Models were colored based on the AlphaFold confidence score (pLDDT) and figures were created using PyMOL (Schrödinger, New York, NY, USA)[90].

**Lysate activity measurements.** Samples were taken from cultures growing on 0.1% (w/v) of a single carbon source or 0.2% (w/v) intracellular enriched bacterial extract. Bacteria were harvested by centrifugation ($4000 \times g$, 10 min) and lysed via sonication on ice in PBS ($3 \times 2$ min, 50% cycle). Debris was removed via centrifugation ($13,000 \times g$, 10 min). Supernatant protein concentration was measured by BSA-assay and 25 µg of protein were incubated with 0.5% of tested polysaccharide for 24 h at RT. Samples containing only polysaccharide or only extract were treated similarly as controls. Lysate activity was measured via DNS-Assay and FACE as described above. Significance of the results was determined via one-way ANOVA followed by post hoc Tukey's HSD test.

**Polysaccharide extraction**
Polysaccharides were extracted from enriched intracellular fractions of *Polaribacter* sp. Hel_I_88. 200 mL culture were harvested via centrifugation ($4000 \times g$, 20 min, 4 °C) and washed once with 10 mL MOPS buffer (20 mM, pH 8) before being resuspended in ddH$_2$O. Polysaccharide extraction was carried out according to a protocol modified from literature[91]. In short, attached particles were removed by centrifugation ($500 \times g$, 10 min) and cells were lysed by sonication on ice ($3 \times 2$ min, 50% cycle). Two more centrifugation steps ($1100 \times g$, 30 min and $27,000 \times g$, 15 min) were carried out to remove unbroken cells and membrane fragments. The pellets were pooled and served as control to ensure enrichment of intracellular polysaccharide (attached fraction). Three volumes of glycine-buffer (0.2 M, pH 10.5) and two volumes of chloroform (both 4 °C) were added to the supernatant and shaken vigorously for 30 s. Phase separation was achieved by centrifugation ($100 \times g$, 2 min) and the aqueous phase removed. The remaining organic phase was re-extracted twice with 2 volumes of glycine buffer and all aqueous phases pooled. They were centrifuged at $47,000 \times g$ for 3 h until a gelatinous pellet remained. After resuspending the pellet in 5 mL ddH$_2$O, 6 volumes of ethanol (4 °C) were added to precipitate the polysaccharides over night. Precipitate was then centrifuged ($14,000 \times g$, 1 h), resolubilized in ddH$_2$O, dialyzed against ddH$_2$O over night to remove residual salts and finally dried in a SpeedVac. Extracts were weighed and stored at −20 °C until further use.

**Bacterial glycan extract characterization**
To determine specific components of the bacterial polysaccharide extracts, 5 mg extract were resuspended in PBS and then were incubated with 25 µg of the characterized enzymes GH13A, GH13B, GH13C and *C. forsetii* GH16[88], respectively. Samples were analyzed by reducing-end assay and FACE as described above. Mono- and oligosaccharide release was compared to samples containing either untreated extract or extract after acid hydrolysis. For acid hydrolysis, 5 mg glycan extract were boiled with 1 M HCl for 2 h, and subsequently neutralized using 1 M NaOH. Monosaccharide composition of all samples was determined via HPAEC-PAD using a Dionex CarboPac PA10 column (Thermo Fisher Scientific) and monosaccharide mixtures as standards[92].

**Determination of glucan concentrations on filters**
Polysaccharide extraction was performed from 3 and 0.2 µm bloom membrane filters of the spring bloom samples and from bacterial single-cultures. Analysis of the extracts was carried out as described[93]. In short, membrane filters were cut into small pieces and extracted using hot ddH$_2$O with sonication treatment and debris was removed via centrifugation ($4500 \times g$, 15 min). α-glucan content was determined via incubation with amylase (*Aspergillus oryzae*, Megazyme) and amyloglucosidase (*Aspergillus niger*, Merck) in sodium acetate buffer (0.1 M, pH 4.5) followed by a PAHBAH-Assay[94]. The measured values were corrected by the amount of water filtered, thereby taking bacterial cell numbers into account.

**Reporting summary**
Further information on research design is available in the Nature Portfolio Reporting Summary linked to this article.

## Data availability
Metagenome, metatranscriptome and MAG sequence data are available from the European Nucleotide Archive (accession PRJEB52999). The mass spectrometry proteomic data have been deposited to the ProteomeXchange Consortium (http://proteomecentral.proteomexchange.org) via the PRIDE partner repository[83] with the dataset identifier PXD019294 (bloom 2016), PXD042676 (bloom 2018), PXD042805 (bloom 2020), PXD043390 (Single strain proteomics). All other data supporting the findings of this study are available within the paper and its Supplementary Information. Source data are provided with this paper.

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

## Acknowledgements

We thank the German Research Foundation (DFG) for funding via the Research Unit FOR 2406 'Proteogenomics of Marine Polysaccharide Utilization' (POMPU) by grants to Uwe T. Bornscheuer (BO 1862/17-3), Jan-Hendrik Hehemann (HE 7217/2-3), Rudolf I. Amann (AM 73/9-3) Hanno Teeling (TE 813/2–3), Mia M. Bengtsson (RI 969/9-2), Dörte Becher (BE 3869/4-3) and Thomas Schweder (SCHW 595/10-3, SCHW 595/11-3). This work was supported by a DFG Heisenberg grant and through the Cluster of Excellence "The Ocean Floor — Earth's Uncharted Interface" project 390741603 to Jan-Hendrik Hehemann. We thank the staff of the Biological Station Helgoland, Alfred-Wegener-Institute Helmholtz Center for Polar- and Marine Research (AWI_BAH_o1) for help with sampling, analyses, logistics, and providing lab space. We thank Michelle Teune for help with TLC analysis and are grateful to Jana Matulla and Tina Trautmann for technical assistance and especially thank Lilly Franzmeyer, Fengqing Wang and Mikkel Schultz-Johansen for environmental sampling and sample processing.

## Author contributions

I Beidler drafted the manuscript, characterized marine enzymes, performed single strain proteomics, prepared and analyzed polysaccharide extracts and performed 18S rRNA sequence analysis. T Schweder and R Amann designed the study and supported the writing of the manuscript together with H Teeling and JH Hehemann. N Steinke and J Krull performed additional glycan extract analysis. T Schulze assisted with growth experiments and enzyme characterization for which T Dutschei provided resources. MK Zühlke and L Torres Martin performed SusD expression and binding studies. Sample measurements for single strain proteomics were performed by B Ferrero-Bordera and J Rielicke. Metaproteome samples were measured by T Sura and V Kale. Mass spectrometry analyses were performed by A Trautwein-Schult. Metaproteome data were analyzed by D Bartosik. C Sidhu performed metatranscriptome analyses and provided data. MM Bengtsson procured 18S rRNA sequence data and performed analyses. K H Wiltshire and I V Kirstein ensured HPLC chl *a* data and provided resources. Figures were prepared by I Beidler and D Bartosik. T Schweder, JH Hehemann, D Becher, UT Bornscheuer, and R Amann coordinated the project. All authors reviewed and approved the manuscript.

## Funding

## Competing interests

The authors declare no competing interests.
