## [Peer Review File · Nature Communications]

Alpha-glucans from bacterial necromass indicate an intra-population loop within the marine carbon cycleReviewer #1 (Remarks to the Author):

The authors should be applauded for the breadth and scope of their work. To take an unexplained observation from the field and come up with a possible explanation down to the role of specific α -glucan related proteins is commendable. This is a great example of the power of collaborative science. While I get the sense from your data that the overall model you outline in Figure 6 will eventually be validated, I am not sure that your data at the time support this model. While not every observation needs to be explained by a mechanism, additional mechanistic insight and data would greatly bolster the authors' model. A number of suggestions to achieve this end are outlined below. Furthermore, should the authors' model be validated, this is of great importance not just for understanding the marine carbon cycle but for the role of α -glucans in other niches like the soil and the gut microbiota.

One reason I don't believe the authors have the data to support their overall model is the apparent lack of any statistical analyses in this manuscript, apart from Figure S4a. I am hoping this is an oversight and do appreciate that with as many authors as are listed here, things get lost. However, this needs to be remedied. Furthermore, averages and standard deviations alone on the graphs are not enough – individual data points need to be displayed throughout.

Figure 1A – I am quite concerned that I can trace the bacterial TCC data in this panel to the gray trace in Figure 1C of PMID: 37069671. Furthermore, the data in Figure 1A and B stop at ~May 20 but the data in C go until May 29. You have the bacterial TCC data until May 29th as this is in your previously published work.

Figure 1D – please indicate how you defined a PUL as an α -glucan one in the legend. Until you have bonafide α -glucan binding data with a representative SusD, I am skeptical of including the SusC/D pairs in this analysis.

Figure 2 – This will be mentioned in my comments about Figure S1, but the authors should consider overexpressing and purifying a couple of representative looped and open SusDs and performing binding experiments to 1) confirm they bind α -glucan and 2) confirm they bind different α -glucans/ α -glucan motifs. Overall, though, Figure 2 was given a lot of space for not a very big point. I might consider putting this in the supplement.

Figure 3 – It does not make sense to root this tree, or include, for that matter, B. theta SusB as this is a GH97, not a GH13. GH13_10s are active on trehalose containing glucans – I would double check CAZy for your annotations here and on all of the GH13s as they have been updated since 2020. Please include more experimental detail in the legend – how long the reaction was performed? What molarity of enzyme? GH13A and C appear to be endo acting (if they can break down β -cyclodextrin) so I am surprised you don't get bigger products (in your FACE analysis) on glycogen. Glycogen is heavily branched and GH13A/C don't seem to accommodate α 1,6 branch points so how is glycogen getting broken down to DP1/2?

Figure 4 – I am surprised by the apparent presence of 70% non-Glc monosaccharides in this intracellular polysaccharide. I would expect this to mostly be bacterial glycogen/ α -glucan. Did you validate that you did not get extracellular polysaccharide in this prep? Either exopolysaccharide or capsule/LPS? In your methods, you mention a gelatinous pellet in line 631 before some final ethanol washes. Glycogen is readily soluble in water which makes me think this gelatinous mixture contains more than just α -glucan. Your methods and statistics in panel B across treatments would be helpful. Panel C – could the authors speculate as to why they get DP2 products with GH13B on this α -glucan but not on glycogen or pullulan in Figure 2? I am not a FACE expert, but why isn't the acid hydrolyzed lane brighter for DP1s?

Figure 5 – Half of your claim in lines 270-272 in the main text are patently false according to your data in Figure 5A. Lysates of *Polaribacter* bacteria grown on α -glucan extract and glycogen appear to have the same activity on glycogen and pullulan. Those grown on alginate do clearly have less activity, however. Without any statistical analyses, I am skeptical of making any conclusions from the data acquired for this Figure. I don't see methods for this experiment. How did you normalize how much protein you used in the DNSA assay. Furthermore, you don't mention what

concentration of polysaccharide the bacteria were grown on for these experiments. Please remedy.

Why is there nearly no activity on glycogen and pullulan in *Muricauda* lysates if the bacteria are apparently able to grow (albeit not well) on glycogen and the α -glucan extract?

Figure 5B – these data don't really match up with the figure from 1A for *Polaribacter* – shouldn't there be more activity on α -glucans if the extract apparently forces the bacteria to make more of the α -glucan PUL proteins than glycogen does? Again – statistics are needed to demonstrate that the extract and glycogen induce more of the α -glucan PUL than alginate. I appreciate that transcription does not equal translation, but I am wondering from a time and cost perspective why the authors chose to do proteomics and not qPCR or even RNAseq? Please update the subfamily annotations in this panel as the GH13 next to the GH65 has been assigned a subfamily and the GH13_10 is probably not a GH13_10.

Supplementary Figure 1 – Panel A. This is misleading – these are annotated as predicted PUL 12 and 13 in the CAZy PULDB, not a single locus as depicted here. Furthermore, part of PUL 12 has been left out of the figure, including another SusC/D pair. Given this is a paper about α -glucans, I would encourage the authors to include GH13 subfamily numbers (updated ones as well). Please also include locus tags on this panel. GH97 enzymes are active on α 1,6 bonds so besides the GH31, this probably contributes to the bacteria's ability to forage on branched α -glucans.

Panel B – it would be helpful to include looped on the left panel label and open on the right panel label, instead of the species names. Please pick a different angle to make your point here, however. The left panel actually looks "looped" to me, either because of the angle or that the modeled cyclodextrin molecule is the same color as the protein. Nonetheless, the right panel looks closed to me. I would encourage the authors to remake this figure with the AlphaFold confidence score colors overlayed onto the structure as the database is notoriously bad at predicting loops. This is also why I am skeptical of making so many conclusions based on a model. The authors have these two isolates in hand and thus I encourage them to overexpress and purify each of these SusD proteins and perform binding studies to confirm that they do bind to an α -glucan ligand.

Panel C – Why did the extract precipitate in the left panel but not the right? I am not familiar with the *Muricauda* species used here – is the growth on extract and glycogen meaningful? Stats are needed to say it grows better on extract/glycogen than pullulan. Could you include MPM only controls (with no carbon source) and a carbon source that it grows well on as a positive control? (This needs to be done for both species, actually). I am suspicious that the *Muricauda* species does not grow well because the only extracellular enzyme part of the PUL in Panel A is a subfamily 38...which you showed has very little activity (that is probably exo). I'm surprised that this organism grew at all, actually, when the GH13_38 showed no activity on glycogen. I would encourage the authors to include a positive control growth condition (glucose?) and do RNAseq to see, compared to growth on glucose, what genes are upregulated by glycogen, pullulan and the extract. Is the entire suite of genes in panel A upregulated when the bacteria grow on α -glucan?

Supplementary Figure 2 – for readers unfamiliar with α -glucans, it might be nice to include diagrams of all the oligo/polysaccharides tested. Please clarify in the legend that the (-) lanes lack enzyme. Please include more detail in the figure legends – like what concentration of enzyme was used and how long the reactions were allowed to proceed. Could the authors explain why they used FACE and not TLC here? It would be helpful to see the non-degraded cyclodextrins and polysaccharides to know they were actually included in the reactions, which you can't see here but can see with TLC.

Supplementary Figure 3C – why are there no CAZymes present in the α -glucan degradation May 2020 bloom? I would also expect some error bars here even if they are only technical replicates.

Supplementary Figure 4 – How can you rule out that the increased abundance of α -glucan was not simply due to the presence of more bacteria? The authors should include growth curves with panel A. Please provide some more detail than "specific enzymatic hydrolyses" in panel B. Which enzymes? Akin to my question with panel A...is there more α -glucan in the later dates simply

because there are more Polaribacter and Aurantivirga? Panel B was also referenced out of order in the main text. Please remedy.

Supplementary Figure 5 – Panel A goes directly against the caption to this figure. I see no increase in abundance over time of Glg proteins. They appear to stay stable from 18 – 48 hours. I see no difference over time in any of the sets of proteins, actually. Please include statistics here to demonstrate the difference, if it exists.

Supplementary Figure 6 – please include more detail in the figure legends about how the lysate reactions were performed.

I am not sure what kind of polysaccharide they extracted. If it's a-glucan then why did they only get 30% glucose from their monosaccharide analysis? How do you know you don't have contaminating extracellular polysaccharide? Is the particular species of Polaribacter you are working with known to make an exopolysaccharide?

Specific lines:

61 – could you define necromass here briefly? This is targeted to a broad audience.

64 – need an "a" between "as" and "major"

87 – PUL, as defined here, is already plural so PULs is redundant. Either say "PUL enable specialized..." Or redefine PUL as polysaccharide utilization locus and then you can say PULs.

87 – what do you mean by "succession"? Did you mean proximity?

125 – you need to cite your previous data here and in the legend for Figure 1.

164 – change "about" to "approximately"

168 – a-glucans should not be plural

187 – please clarify where these "isolates" came from.

190 – this is mentioned elsewhere but if you have GH97s highly represented in your isolates, that could be the source of a1,6 activity. Do you have a citation for the GH13_31 a1,6 activity? Can you purify the enzyme and show this?

197 – Glycogen and pullulan contain both a1,4 and a1,6 linkages. Glycogen is not mostly a-1,4-linked. It contains a significant number of a1,6 branch points, more than plant amylopectin.
<https://doi.org/10.1016/B978-0-12-821848-8.00172-4>

199 – Please clarify that this product is likely panose otherwise it seems like this enzyme is a dedicated pullulanase that hydrolyzes a1,6 bonds to produce maltotriose.

216 – until you purify an enzyme with a1,6 activity, I don't think you can say the marine bacteria are adapted towards glucan containing a1,6 bonds.

224 – "linear glycogen" is a bit of a misnomer. The definition of glycogen is that it is a branched molecule. Maybe call it a1,4 glucan at this point.

231 – "laminarin-targeting Glg-proteins" is a bit confusing. Could you clarify what you meant here?

233 – Is this really a response specific to only laminarin? Would growth on alginate or fucoidan lead to a-glucan synthesis?

234 – need a "the" before "sole"

237 – same question as in line 233

257 – Since this α -glucan storage and release is central to your model, I think it is imperative for you to run some analytical techniques, like NMR, on this purified polysaccharide to determine its structure. Do you simply have bacterial glycogen or is there something special about *Flavobacteriia* intracellular α -glucan?

277 – I think you meant to cite 5B and not 5A. I would recommend doing qPCR and not proteomics to show this.

288 – The difference in α -glucan utilization you observe is likely due to the presence of different GH13s, not the SusDs. If you wish to make the point that SusDs also drive this difference, they need to be expressed, purified and have their α -glucan binding ability characterized.

299 – I don't think it is fair to say that the *Muricauda* strain can "quickly take up and recycle these α -glucans" when it didn't show very robust growth on extracted α -glucan, pullulan or glycogen.

310 – *Muricauda* targeting simple alpha glucans is a bit confusing to me. By simple do you mean shorter or just less complex/branched? Do you think *Muricauda* grows on α -glucan polysaccharide degraded by *Polaribacter*? The *Polaribacter* enzymes kind of go against the "selfish PUL" paradigm if they are capable of degrading α -glucan to DP1/2 and they contain SP2 signal peptides (ie are all on the outside of the cell). Do you think *Muricauda* forages on these DP1/2 products?

327 – I don't think it is right to say they are structurally similar when you haven't characterized the structure of your proposed bacterial α -glucan.

351 - do you have a reference for this statement?

376-78 – Is it fair to compare the extracted pigments if they were purified differently?

412 – stated elsewhere but please update predicted GH13 subfamilies to the current CAZy database.

432 - -1 needs to be in superscript.

550 – Please list the source of all oligo and polysaccharides used in your work.

587-589 - Your methods describe cloning of GH13 proteins 1, 2 and 3 but the main text describes them as A, B and C. Please be consistent. Were these cloned without the predicted SP2 signal peptides?

604 – please also provide final protein concentrations. 25 ug is a lot of protein....especially to let the reaction run for 24 hours.

633 – how did you validate that you didn't have any membrane bound polysaccharides?

637 – stated elsewhere but it would be standard practice to characterize your polysaccharide via an initial isoamylase treatment (to remove branch points and allow you to determine the degree of branching and chain length distribution) followed by β -amylase/ α -amylase treatment. See PMC7407607

Minor: the authors provide gene locus tags that don't correlate with protein locus tags. Could the authors provide protein names and what database they are derived from (GenBank, IMG, UniProt?).

Reviewer #2 (Remarks to the Author):

Beidler et al describe an investigation into glucose cycling in the ocean, by means of bacterial α -glucans that appear to be recycled by the bacterioplankton community. With algal laminarin serving as primary input carbon source, bacteria produce α -glucans, undergo cell lysis, and the released α -glucans are taken up by other species, forming a loop. Support is drawn from genomic analyses, growth studies, transcriptomics, proteomics, and enzyme characterisation work. The story is very compelling, and the work represents a major advance, building on previous insightful studies from this cluster of authors looking at the importance of laminarin in the ocean. This work defines an aspect of carbon cycling that had previously been overlooked, and indeed establishes bacterial glycans as a major carbon source for bacterioplankton. While the overall story told in the paper is very compelling, some of the experimental data are not quite convincing in their support of specific points of discussion. Some data points may be over-interpreted when drawing conclusions. And the writing could be improved in places, for better clarity. The points below should be addressed prior to publication, but on the whole this is a very nice and sound piece of work.

Some references are out of date. Reference 10 is a review from 2009, cited in the Introduction as a general discussion on PULs, a topic on which many more reviews have been published in the past 3-5 years. Moreover, the review cited focusses on human gut symbionts, whereas other reviews have been published that also discuss environmental species and may therefore be more relevant. Similarly, reference 57 for the CAZy database is very out of date (2009) and has been superseded more than once. See the CAZy website for guidelines on how to cite the use of the tool today (i.e., there is a paper in *Nucleic Acids Res* from 2022).

In the Introduction on pg5, the authors say "In this study, we show that marine Flavobacteria...degrade the microalgal β -glucan laminarin and use excess glucose to synthesize α -glucan storage polysaccharides." Where is the evidence in this paper that excess non-metabolised glucose from laminarin goes directly into a pathway for α -glucan synthesis? A similar statement is made in the Discussion on pg13, "Our data suggest that a substantial fraction of this laminarin is not immediately remineralized but rather converted to bacterial α -glucans." While this is an interesting theory and would fit with the data presented, I don't agree that it has been proven or even tested in this study. The authors do verify that "laminarin degradation coupled with simultaneous α -glucan-synthesis could be confirmed with an isolated strain in vitro" but I don't agree that these data show a direct transfer of glucose from metabolised laminarin into an α -glucan synthesis pathway.

The "Helgoland Roads time series" is first mentioned on line 137 and is not defined – please provide some context as to what this is/was.

On line 140 the authors state that "Dinophyceae also known to contain α -glucans...made up 7 to 47% of the eukaryotic population." Since the abundance of some of these species correlates well with the 'main phase' second bloom, why were they not considered another major source of α -glucan substrate, which would disrupt the notion of an almost self-sustaining bacterial loop?

In the paragraph beginning line 155, it would be helpful to mention the enzyme specificities typically found for GH13 enzymes, as not all readers will be familiar with the family.

Lines 163-164, can proteins be classed into functionally distinct groups based only on structure predictions, without any functional characterisation data? The proposed differences in specificity make sense but it appears that the GH13 abundance in PULs with different types of SusD is the only supporting evidence for a difference in binding specificity, since both SusDs seem to have been modelled with the same ligand.

Line 172, a small issue of lack of clarity in writing. Is "more GH13s" really an indicator of greater diversity than "some GH13s" ?

In the growth studies shown in Figure S1B-C, I would like to see control experiments using glucose and a "no carbon source" unsupplemented medium to see the baseline for minimal possible growth. *Polaribacter* growth on the extract is minimal (and apparently confounded by aggregation issues), and growth for *Muricauda* is at the same minimal level for all experiments shown here. Given the overall low level of growth for *Muricauda*, I don't support the statement on line 181 of the main manuscript that there is evidence of a 'strong' preference for glycogen versus 'poor growth' on pullulan, since both seem poor. A glucose control experiment would be very helpful for demonstrating 'good' growth, and an experiment without any carbon source would clarify whether either species is able to use the extract at all.

On Figure S1B, consider highlighting (where present) the loop that is absent in the 'open' SusD structure.

The paragraph on line 186-192 is very unclear, please try to re-phrase all of it. I am not sure what was compared to what, or for what purpose.

The three characterised GH13 enzymes are referred to in text as GH13A, GH13B, and GH13C, starting on pg8. Yet the PUL shown in Figure 5 does not indicate which these are in the PUL. And in the Methods section on pg23 (Cloning section) and pg25 (Bacterial glycan characterisation section) they are referred to as GH13 (1), GH13 (2), and GH13 (3). Confusing! Naming consistency is essential.

The FACE results are not always clear, with some showing a "simile/frown" artefact that makes comparison with standards difficult, and many showing a green line across all or most lanes. Since HPAEC-PAD was used for other experiments, why was it not used for oligosaccharide analysis? The data could have been cleaner.

On pg9, line 231, what are the "laminarin-targeting" glg proteins? This paragraph in general is rather unclear and confusing: since Glg proteins are presented as being involved in α -glucan synthesis, whereas "laminarin-targeting" implies degradation. On Figure S3, panel B is labelled with " α -glucan biosynthesis" and in the legend it is described as showing "Glg-pathway" – how does that relate to laminarin?

On line 244 it states "...we showed increased amounts of α -glucan present in the culture (Fig S5C)." But Figure S5C shows protein abundance of α -glucan PUL proteins. Where is the data showing the amount of α -glucan accumulating in culture medium?

More detail is needed about the characterisation of the extract. Figure 4A shows sugar composition of the extract, and the legend implies that this was achieved using enzyme hydrolysis – is that accurate? The Methods section would suggest acid hydrolysis was used, and I assume this was for analysis of the extract, so the Figure 4 legend should be corrected. In addition, how was the "mol%" content of each sugar determined? Is this representative of the entire sample, or only of the carbohydrate portion of the sample? (i.e. were the amounts of each sugar detected by HPAEC simply added together, or was a % of sample original weight determined?) The purity of the polysaccharide in terms of the absence of protein, lipid, etc is very important in understanding results of the growth study.

Standards are not labelled on the FACE image shown in Figure 4C.

Lines 273-274, where protein abundance is compared to alginate controls. Some of the proteins from the PUL seem quite abundant in the alginate sample, while others do not – how do the authors account for this, and was alginate the most suitable choice as a control here?

Please check your references to Fig 5A and Fig5B – panel A shows enzyme assays for both species, panel B shows protein abundance for both species. (Mis-referenced on line 280 and possibly elsewhere)

Lines 280-282 seem to assume there are no α -glucanases from outside the PUL that are being expressed, have the authors looked at the whole genome to verify this?

The paragraph in lines 299-307 says that the authors "have shown" that α -glucan uptake is "quick" and recycling is "rapid" – where has this been tested in this study? Have the authors for example examined the speed of transcriptional response in the PULs? I don't see such supporting data.

Line 319, did the authors consider characterising this presumed α -1,6- specific GH13_31? It would give clear support to the notion of a PUL targeting more complex polysaccharides.

In the Methods, I note that the CAZy database was accessed in 2020 for transcript annotation. The database has been updated several times since then, with new families and sub-families being defined very frequently. Is it feasible to re-annotate the data with a more recent database?

Line 551-552 – why was a higher concentration of the extract provided compared to other polysaccharides, and does this not skew the growth data?

Line 569 – I assume (brand) is a holdover from an unfinished version of the manuscript.

What was the reaction volume used in enzyme characterisation assays (bottom of pg23)? And on line 611-612, what does it mean that assays were compared to solutions with only enzyme OR polysaccharide? Which was used as baseline?

A GH16 enzyme from *Christiangramia forsetii* is mentioned in a figure legend, but it is not referred to anywhere in the text. There should be explanation somewhere of why this was used.

Reviewer #3 (Remarks to the Author):

General comments:

Beidler and colleagues combine field and culture data to assess the role of laminarin degradation products in fueling secondary production in marine bloom events. They use proteomics, carbohydrate analyses, culture experiments and recombinant enzyme characterization to identify the proteins and linkage patterns associated with bacterial usage of algal-derived carbohydrates such as laminarin. The authors are quite thorough in their analyses and pose a compelling set of interpretations from their data. However, my impression is that their overall point is not necessarily supported by the data that they present. The data thoroughly covers the production and consumption of α -glucans by bacteria and their specificity towards different glucan linkages – but it does not cover the field component which is then summarized in Figure 6. Specifically, the temporal evolution of the system is not well-supported by the data presented here. The data is great, but I wonder if it is being used to tell the wrong story, or at least to speculate further than the data can go? For example, in Figure 6 (the summary figure), there are a few places where data seems to be missing and/or incomplete. Is there clear evidence of the temporal separation between steps 3 and 5? My read of the data presented is that the progression from B-glucan PULs to α -glucan PULs is not available here. If that is wrong, perhaps this data could be displayed or described more prominently. Second, it is not clear if the soluble α -glucans depicted in the center of the figure have been observed in seawater by the authors. Certainly glucose and its small polymers are vanishingly dilute in seawater, but do the authors have data on the rates of turnover of this pool? I believe that the turnover rates could be quite fast, but there is no data to constrain these values and test this hypothesis. Finally, the grazing / viral lysis hypothesis for α -glucan production is not tested. There has been some nice work showing that intracellular composition is different between viral-infected and non-infected cells. Can the authors assess whether (or how much) this affects their hypothesis of lysed cells releasing α -glucans into the dissolved pool? Would

infected and non-infected cells provide the same approximate amount of α -glucans to the dissolved pool? Does anyone know the answer to this question?

In general, then, I am impressed by the work represented in this paper, but I wonder if the authors have stretched the data a little further than is merited. I provide some additional specific comments below to improve the manuscript.

Specific comments:

1. Page 13, paragraph starting Line 331: I am not following the calculations presented here as to the importance of this process for bacterial cycling in marine systems. If these compounds are degradation products of laminarin, then isn't their carbon content accounted for in the laminarin value? My sense of this calculation is that it is based on cellular content prior to release and degradation. I'm not sure that these degradation products constitute an additional pool of carbon. I do think that the rate of degradation is important, particularly if the degradation steps release CO₂ and thus they could constitute an important component of respiration. But the data and calculations presented here do not touch on this topic and so I am not clear on how this calculation helps us consider carbon cycle dynamics in a new way.
2. Figure 1 caption: I find the figure captions to be challenging for a non-expert to interpret so I am providing some areas of confusion for myself in the hope that fixing them will make the figures more accessible to a more general audience. For Figures 1C and 1D, why are there no values given for the y-axis? I realize that these are stacked figures, but could there be an example (or representative) y-axis for one of the graphs. Some sense of the range would be helpful for both C and D.
3. Figure 2 caption: The manuscript text referencing this figure discusses GH13 and other enzymes, but then they are not depicted anywhere in the figure. Can this be done? I am not an expert in these figures, so I can't make sense of where GH13 would fall in this tree and thus I do not have a good reference point for the figure.
4. Figure 5 caption: In A, please describe the different bars in the graph. How does "release equivalent (glucose)" mean in the context of lysate activity (the only descriptor for Figure 5A)? For B, what is the y-axis? (what is riBAQ? This abbreviation is not described in the caption.)

Point-to-point reply to the reviewers' comments

Manuscript NCOMMS-23-35735A | Alpha-glucans from bacterial necromass indicate an intra-population loop within the marine carbon cycle | Beidler *et al.*

We are grateful for the rigorous review of our manuscript and the many useful comments and suggestions that were made by the reviewers including scientific clarifications and reference suggestions, which helped us to considerably improve this study. We have taken all points of critique that were raised by the reviewers into account, performed additional experiments and revised the manuscript accordingly, as we detail below.

Reviewer #1

The authors should be applauded for the breadth and scope of their work. To take an unexplained observation from the field and come up with a possible explanation down to the role of specific a-glucan related proteins is commendable. This is a great example of the power of collaborative science. While I get the sense from your data that the overall model you outline in Figure 6 will eventually be validated, I am not sure that your data at the time support this model. While not every observation needs to be explained by a mechanism, additional mechanistic insight and data would greatly bolster the authors' model. A number of suggestions to achieve this end are outlined below. Furthermore, should the authors' model be validated, this is of great importance not just for understanding the marine carbon cycle but for the role of a-glucans in other niches like the soil and the gut microbiota.

We thank the reviewer for the positive feedback on our work and appreciate the constructive criticism. We have revised the core message of the manuscript including the model shown in Figure 7 (previously Figure 6) and have shifted the focus more towards the previously overlooked bacterial alpha-glucan loop. Furthermore, we have performed additional experiments and added statistics for all applicable analyses to support our hypothesis. We believe that these improvements have resulted in a more clear and better supported presentation of our work.

One reason I don't believe the authors have the data to support their overall model is the apparent lack of any statistical analyses in this manuscript, apart from Figure S4a. I am hoping this is an oversight and do appreciate that with as many authors as are listed here, things get lost. However, this needs to be remedied. Furthermore, averages and standard deviations alone on the graphs are not enough – individual data points need to be displayed throughout.

We concur and have modified the data presentation accordingly. Statistical analyses for all experiments were performed and are now listed in the manuscript as well as depicted in the figures where possible. Individual data points with standard deviations and mean values are now also given whenever applicable.

Figure 1A – I am quite concerned that I can trace the bacterial TCC data in this panel to the gray

trace in Figure 1C of PMID: 37069671. Furthermore, the data in Figure 1A and B stop at ~May 20 but the data in C go until May 29. You have the bacterial TCC data until May 29th as this is in your previously published work.

We have modified Figure 1. TCC data, which now includes data up to 29 May. These data are indeed the same as those we published in PMID: 37069671, as this work is based on observations we made during the exact same 2020 Helgoland phytoplankton bloom. We believe that in order to illustrate the overall dynamics of the 2020 spring bloom, it is necessary to include some of the previously published data as a reference point for readers of this paper. We now indicate in the legend that these overview data of the bloom situation in 2020 have already been shown in PMID: 37069671, whereas the 18S rRNA gene analysis is new.

Figure 1D – please indicate how you defined a PUL as an a-glucan one in the legend. Until you have bonafide a-glucan binding data with a representative SusD, I am skeptical of including the SusC/D pairs in this analysis.

We appreciate this comment and have specified in the Methods section how we identified and assigned putative substrates to PULs obtained from high-quality MAG data based on our previously published PUL-calling methods, which evaluate the positioning of validly annotated CAZymes with a SusC/D pair in close proximity. Even without bonafide binding data, previous work (e.g. PMID: 30111868, PMID: 31316134, PMID: 33649555) strongly indicates that SusC/Ds associated with specific substrates show a high degree of sequence conservation and are often co-localized with specific CAZymes within a PUL, which we see similarly for the SusC/Ds included in our analyses.

This is also confirmed by a significant up-regulation of PUL-associated SusC/D proteins when bacterial isolates were grown on predicted PUL-specific polysaccharides (e.g. PMID: 30111868, PMID: 37121608, PMID: 36411326, PMID: 31285597). This substrate-specific co-regulation of the susC/D genes could also be demonstrated in this study with our model strains *Polaribacter* sp. and *Muricauda* sp. grown on alpha-glucans. As shown in detail in Tab. S4, these SusC/D proteins are the only PUL-associated SusC/D proteins highly expressed in these cultures, each being among the 20 most abundant proteins of the samples. Furthermore, they are the only SusC/Ds of these cultures that show a high fold change compared to cultures grown on different polysaccharides. Furthermore, the binding studies now presented in Figure S2 confirm that the looped alpha-glucan SusD proteins bind alpha-glucans such as pullulan and glycogen.

Figure 2 – This will be mentioned in my comments about Figure S1, but the authors should consider overexpressing and purifying a couple of representative looped and open SusDs and performing binding experiments to 1) confirm they bind a-glucan and 2) confirm they bind different a-glucans/a-glucan motifs. Overall, though, Figure 2 was given a lot of space for not a very big point. I might consider putting this in the supplement.

We cloned and overexpressed a couple of SusD-encoding genes of both alpha-glucan PUL types (e.g., from *Polaribacter* sp., *Muricauda* sp. and *Flavimarina* sp.). We were lucky to get a functional expression of the alpha-glucan PUL-encoded SusD of *Flavimarina* sp. Hel_I_48 (looped group). We now present an additional Figure S2, showing retardation of this SusD from *Flavimarina* sp. Hel_I_48 in a native gel containing the alpha-glucans pullulan and glycogen but not laminarin,

supporting substrate-specificity towards alpha-glucans. It is worth mentioning that the surface model of this a-glucan specific SusD from *Flavimarina* sp. Hel_I_48 (see Figure S3) is almost identical to the looped SusD structure of the protein from *Polaribacter* sp. Hel_I_88.

In addition to these results, we rely on the presented model further detailed in Figure S3, which has also been revised for clarity. The text has additionally been modified to reflect the fact that in “looped” marine alpha-glucan PUL SusDs, most of the binding residues characterized in *B. thetaiotaomicron* SusD are conserved, strongly indicating a similar substrate and binding mechanism. The residues missing in the “open” marine alpha-glucan PUL SusDs are both located on the missing loop, again indicating that it plays a role in substrate binding and that “open” marine alpha-glucan PUL SusDs may employ a different mode of binding. Furthermore, the strong upregulation of the looped alpha-glucan PUL SusD in proteome experiments performed with strains grown on alpha-glucan is a strong indication that they are responsible for the alpha-glucan binding and uptake of these substrates. Of course, these findings are implications that the specific mode of binding for both types of PULs needs to be further investigated as part of a follow-up study.

Figure 3 – It does not make sense to root this tree, or include, for that matter, *B. theta* SusB as this is a GH97, not a GH13. GH13_10s are active on trehalose containing glucans – I would double check CAZy for your annotations here and on all of the GH13s as they have been updated since 2020. Please include more experimental detail in the legend – how long the reaction was performed? What molarity of enzyme? GH13A and C appear to be endo acting (if they can break down b-cyclodextrin) so I am surprised you don't get bigger products (in your FACE analysis) on glycogen. Glycogen is heavily branched and GH13A/C don't seem to accommodate a1,6 branch points so how is glycogen getting broken down to DP1/2?

We appreciate this comment and have removed *B. thetaiotaomicron* SusB from the tree and now root it at mid-point.

Thank you for pointing out the updated version of the CAZy database. We initially used the database that was current at the time of manuscript submission (CAZy V11). In the revised manuscript, we have used the new version that has since been released and have therefore updated our annotations.

Selected experimental details on how long reactions with the GH13s were performed and how much enzyme was used, as well as all other specifications, are now also briefly mentioned in the description of Figure 3. Please note that a detailed description of these experiments can be found in the Methods section.

Regarding product size - it may be that our endpoint analysis prevented us from observing larger oligosaccharides that might initially be released. It can be assumed that analyzing samples from different time points would show that the product becomes smaller over time.

Figure 4 – I am surprised by the apparent presence of 70% non-Glc monosaccharides in this intracellular polysaccharide. I would expect this to mostly be bacterial glycogen/a-glucan. Did you validate that you did not get extracellular polysaccharide in this prep? Either exopolysaccharide or capsule/LPS? In your methods, you mention a gelatinous pellet in line 631 before some final

ethanol washes. Glycogen is readily soluble in water which makes me think this gelatinous mixture contains more than just α -glucan. Your methods and statistics in panel B across treatments would be helpful. Panel C – could the authors speculate as to why they get DP2 products with GH13B on this α -glucan but not on glycogen or pullulan in Figure 2? I am not a FACE expert, but why isn't the acid hydrolyzed lane brighter for DP1s?

We agree that presence of ~70% non-glc monosaccharides in our extracts could be caused by an imperfect separation. Indeed, we performed a control experiment involving the pooled fractions taken after the initial centrifugation steps described in our methods section (pre- and post-sonication), and the analysis of this extract included higher amounts of galactosamine, so we assume there is a degree of separation occurring via our extraction method. We now include this analysis in Figure 5 (previously Figure 4). Based on these results, we cannot guarantee that our extract does not contain a certain amount of exopolysaccharide contamination, but the experiments performed with our enzymes prove that the extract contains significant amounts of α -glucan. When compared to *Polaribacter* sp. exopolysaccharide preparations (PMID: 16052372), our extract shows a much higher Glc%, leading us to believe that we have at least enriched intracellular bacterial polysaccharide.

We deliberately chose an extraction method derived from heavy matrix-producing bacteria to minimize contamination. As for the gelatinous pellet, the pellet was also observed in the extraction of high-percentage glucose containing extracts, so the formation of the pellet alone does not indicate presence or absence of other monosaccharides. As our experiments with the GH13s gave specific activities, we concluded that our method for an enrichment of α -glucan extracts was successful.

Statistics on the data are now included in the figure.

Panel C – We indeed observed the same DP2 band when incubating either pullulan or glycogen with GH13B as when incubating the extract with the enzyme, as can be seen in Figure S5 and in the significant increase in reducing sugar release depicted in Figure 3.

Figure 5 – Half of your claim in lines 270-272 in the main text are patently false according to your data in Figure 5A. Lysates of *Polaribacter* bacteria grown on α -glucan extract and glycogen appear to have the same activity on glycogen and pullulan. Those grown on alginate do clearly have less activity, however. Without any statistical analyses, I am skeptical of making any conclusions from the data acquired for this Figure. I don't see methods for this experiment. How did you normalize how much protein you used in the DNSA assay. Furthermore, you don't mention what concentration of polysaccharide the bacteria were grown on for these experiments. Please remedy.

We apologize for this and have adapted the mentioned lines to better reflect the data. Likewise, methods for this experiment have been added to the Methods section. Statistics are now included and prove significantly higher activities of culture lysates when grown on extract or glycogen compared to controls as well as an increase in the abundance of α -glucan-targeting proteins. We have added the polysaccharide concentration to the legend of Figure 6 (previously Figure 5). Further details of how the strains were cultivated can be found under "Strain and cultivation conditions" in the Methods section.

Why is there nearly no activity on glycogen and pullulan in *Muricauda* lysates if the bacteria are apparently able to grow (albeit not well) on glycogen and the α -glucan extract?

As we now show with the added statistics, lysate from a *Muricauda* culture grown on either extract or glycogen revealed a significantly increased activity on different α -glucans compared to a control.

Figure 5B – these data don't really match up with the figure from 1A for *Polaribacter* – shouldn't there be more activity on α -glucans if the extract apparently forces the bacteria to make more of the α -glucan PUL proteins than glycogen does? Again – statistics are needed to demonstrate that the extract and glycogen induce more of the α -glucan PUL than alginate. I appreciate that transcription does not equal translation, but I am wondering from a time and cost perspective why the authors chose to do proteomics and not qPCR or even RNAseq? Please update the subfamily annotations in this panel as the GH13 next to the GH65 has been assigned a subfamily and the GH13_10 is probably not a GH13_10.

Regarding RNAseq – as the reviewer already points out, transcription does not equal translation. We therefore decided to use more time-consuming proteomics experiments to uncover increases or decreases in protein levels which can more readily support our hypothesis.

Statistics are now included in the figure and clearly demonstrate not only that α -glucan-based growth significantly induces the expression of most PUL-encoded proteins, but also that growth on our extract leads to higher induction of gene expression compared to growth on glycogen.

The lack of a significant difference in lysate activity between glycogen- and extract-grown cultures is probably due to the fact that our reducing end assay is an end-point measurement that does not provide any kinetic information. From our proteomics data it can be assumed that in a reducing end assay with multiple time points measured, the lysate from the extract-grown culture would yield higher values more quickly than a glycogen-based culture. However, due to lysate volume limitations, we opted for an end-point analysis.

As the method section details, CAZyme annotations of isolates were previously based on dbCAN-HMMdb-V11 searches, as it was the most recent database at the time of submission. We appreciate that using the version that has since been released is now more appropriate and thus annotations were updated to dbCAN-HMMdb-V12. However, the mentioned GH13_10 remains as such. This hit could not be confirmed using CAZyDB and is thus marked with an asterisk in the annotation tables (Table S2-4).

Supplementary Figure 1 – Panel A. This is misleading – these are annotated as predicted PUL 12 and 13 in the CAZy PULDB, not a single locus as depicted here. Furthermore, part of PUL 12 has been left out of the figure, including another SusC/D pair. Given this is a paper about α -glucans, I would encourage the authors to include GH13 subfamily numbers (updated ones as well). Please also include locus tags on this panel. GH97 enzymes are active on α 1,6 bonds so besides the GH31, this probably contributes to the bacteria's ability to forage on branched α -glucans.

Figure S1A has been updated for more clarity and now includes locus tags as well as subfamily annotations. As we use different PUL calling (explained above) than PULDB, the second SusC/D pair has been deliberately omitted as it is most likely part of another PUL downstream that seems to target beta-glucan. In our proteomics datasets (Tables S3/S4) we observe much higher (100-1000-fold) protein amounts for this SusC/D pair when grown on laminarin compared to either our extract or glycogen and therefore conclude that it is not part of the same regulon.

Panel B – it would be helpful to include looped on the left panel label and open on the right panel label, instead of the species names. Please pick a different angle to make your point here, however. The left panel actually looks “looped” to me, either because of the angle or that the modeled cyclodextrin molecule is the same color as the protein. Nonetheless, the right panel looks closed to me. I would encourage the authors to remake this figure with the AlphaFold confidence score colors overlaid onto the structure as the database is notoriously bad at predicting loops. This is also why I am skeptical of making so many conclusions based on a model. The authors have these two isolates in hand and thus I encourage them to overexpress and purify each of these SusD proteins and perform binding studies to confirm that they do bind to an α -glucan ligand.

The angle on Figure S1B (now S3) has been updated to better showcase the different conformations of our SusD models. Confidence scores have been included. We appreciate the point about AlphaFold being bad at predicting loops, but as can be seen in Figure S3, the mentioned loop has been predicted by AlphaFold with “high” confidence score. Additionally, the fact that all *B. theta* - like “looped” SusDs contain a ~14 amino acid long conserved extension that includes a residue characterized as binding in *B. thetaiotaomicron* is a strong indication of a similar binding mechanism. The initial work done on *B. theta* SusD also indicated that this loop is flexible since it was not observed with the solved structure. All binding residues characterized in *B. thetaiotaomicron* are conserved in looped SusDs, yet the absence of the loop in the open SusDs accounts for two missing binding residues. As is now included with Figure S2, we show binding for the looped-group *Flavimarina* sp. alpha-glucan PUL SusD type to alpha glucans glycogen and pullulan but not laminarin.

Unfortunately, our attempts to obtain soluble, correctly folded SusDs for detailed binding studies with open-type SusD proteins have so far not been successful. However, our PUL-calling indicates both SusDs are conserved parts of different alpha-glucan targeting PULs, and their structural differences strongly indicate distinct binding mechanisms – which of course need to be further explored.

Panel C – Why did the extract precipitate in the left panel but not the right? I am not familiar with the *Muricauda* species used here – is the growth on extract and glycogen meaningful? Stats are needed to say it grows better on extract/glycogen than pullulan. Could you include MPM only controls (with no carbon source) and a carbon source that it grows well on as a positive control? (This needs to be done for both species, actually). I am suspicious that the *Muricauda* species does not grow well because the only extracellular enzyme part of the PUL in Panel A is a subfamily 38....which you showed has very little activity (that is probably exo). I’m surprised that this organism grew at all, actually, when the GH13_38 showed no activity on glycogen. I would encourage the authors to include a positive control growth condition (glucose?) and do RNAseq to see, compared

to growth on glucose, what genes are upregulated by glycogen, pullulan and the extract. Is the entire suite of genes in panel A upregulated when the bacteria grow on α -glucan?

We have revised this figure and its description for clarity. It was the bacterial population that formed aggregates, not the extract. This is a typical behavior we have observed with many of our marine isolates – some substrates cause the bacteria to form aggregates, others do not. MPM negative and glucose positive controls are now included. While our data shows that the strains grow by far the best on glucose, growth on glycogen and extract is still relevant. The behavior of growing well on monosaccharides – yet less well on the corresponding polysaccharide – while still exhibiting significant activity, has been previously observed for this strain (PMID: 36411326).

Our proteomic analyses of the *Muricauda* sp. isolate (Table S4) did not show any significant up-regulation in other PULs. Since GH13_38 showed activity on the extract as well as on glycogen and on pullulan (albeit low activity, Figure S5B, previously S2B), this may help explain the stunted growth on these polysaccharides. Proteins encoded within the α -glucan PUL of *Muricauda* sp. show a continuously high expression, regardless of the substrate on which they grow. This could indicate a different regulation and different utilization mechanism, which will be the subject of follow-up studies.

Supplementary Figure 2 – for readers unfamiliar with α -glucans, it might be nice to include diagrams of all the oligo/polysaccharides tested. Please clarify in the legend that the (-) lanes lack enzyme. Please include more detail in the figure legends – like what concentration of enzyme was used and how long the reactions were allowed to proceed. Could the authors explain why they used FACE and not TLC here? It would be helpful to see the non-degraded cyclodextrins and polysaccharides to know they were actually included in the reactions, which you can't see here but can see with TLC.

The figure legend was adapted accordingly. In addition to our FACE-analysis, we now include TLC analysis of the tested cyclodextrins and polysaccharides where the non-degraded substrate wasn't visible in FACE (Fig. S5). These results confirm those of our FACE experiments and clearly show which cyclodextrins/polysaccharides are targeted by our tested enzymes.

Supplementary Figure 3C – why are there no CAZymes present in the α -glucan degradation May 2020 bloom? I would also expect some error bars here even if they are only technical replicates.

We acknowledge that the bars are quite small, but we do see CAZymes targeting α -glucans in the bacterial microbiome of the phytoplankton bloom during May 2020. CAZymes in general are usually detected at low levels for all substrates, which is why previous studies (Krüger et al. PMID: 31316134, Francis et al. PMID: 33649555) have focused heavily on SusC/D proteins, as they are by far the most abundant PUL specific marker proteins. Metaproteomics always has a bias towards detecting the most abundant protein groups, which is why seeing α -glucan targeting CAZymes at all is an indication of importance. We do however appreciate that the view obtained from metaproteomics is quite general and therefore we now also include the new Figure 4, in which we show metatranscriptomics data from the phytoplankton bloom of 2020, analogous to Figure S6, resolved to genus level, confirming what could be seen in metaproteomics.

In metaproteomics, we usually calculate the abundance of proteins (groups) relative to the whole. Average values of identified proteins were calculated from three biological replicates, including '0' for proteins that were not identified within a replicate. As soon as the measured quantitative values (already normalized to the respective protein size) per protein are divided by the sum of all measured quantitative values of the sample, it is no longer possible to calculate error bars for individual proteins. In addition, for the data shown here, the %NWS of individual proteins are summed according to their functional classification (e.g., CAZyme families). Please also note that we only include proteins in this dataset if they have been unambiguously identified with at least two peptide identifications per protein sequence in at least two biological replicates.

Supplementary Figure 4 – How can you rule out that the increased abundance of α -glucan was not simply due to the presence of more bacteria? The authors should include growth curves with panel A. Please provide some more detail than “specific enzymatic hydrolyses” in panel B. Which enzymes? Akin to my question with panel A...is there more α -glucan in the later dates simply because there are more *Polaribacter* and *Aurantivirga*? Panel B was also referenced out of order in the main text. Please remedy.

We have revised the Methods section to better reflect how we processed these samples and how the data was obtained. The values shown are corrected for the amount of water filtered, i.e., the more bacteria are present, the less water is filtered. Therefore, the values for both panels A and B take into account the changing number of bacteria.

In addition, as shown in Figure 1A, bacterial cell numbers are not highest during the peak bloom phases, but actually around mid-April. The only bacterial group that responded strongly to the diatom bloom were the *Flavobacteriia*, and their relative abundance increased rapidly during the peak bloom phases, such as in early May 2020. See also PMID: 37069671 for relative abundance by clade during the 2020 bloom.

We now detail the enzymes used for hydrolysis in the figure description. This information can also be found in the Methods section.

The out-of-order reference was corrected.

Supplementary Figure 5 – Panel A goes directly against the caption to this figure. I see no increase in abundance over time of Glg proteins. They appear to stay stable from 18 – 48 hours. I see no difference over time in any of the sets of proteins, actually. Please include statistics here to demonstrate the difference, if it exists.

The caption of the Figure S5 (now S8) does not indicate an increase of any proteins and the text clearly states that “proteins encoded by the *glg*-operon were expressed continuously during growth on laminarin”. Statistics are now included and do indeed reveal a significant increase for GlgE over the course of the experiment.

Supplementary Figure 6 – please include more detail in the figure legends about how the lysate reactions were performed.

Details on how reactions were performed are now briefly mentioned in the figure caption. They can be found in more detail in the Methods section.

I am not sure what kind of polysaccharide they extracted. If it's a-glucan then why did they only get 30% glucose from their monosaccharide analysis? How do you know you don't have contaminating extracellular polysaccharide? Is the particular species of *Polaribacter* you are working with known to make an exopolysaccharide?

Please see the previous comments regarding the purity of our extract.

Specific lines:

61 – could you define necromass here briefly? This is targeted to a broad audience.

Necromass is now defined in the abstract and the introduction.

64 – need an “a” between “as” and “major”

Corrected.

87 – PUL, as defined here, is already plural so PULs is redundant. Either say “PUL enable specialized....” Or redefine PUL as polysaccharide utilization locus and then you can say PULs.

This has been adapted accordingly.

87 – what do you mean by “succession”? Did you mean proximity?

To avoid confusion, we now term it “temporal succession”.

125 – you need to cite your previous data here and in the legend for Figure 1.

We have now added this information and cite our recent paper (PMID: 37069671) in the text and the figure legend.

164 – change “about” to “approximately”

This was corrected.

168 – a-glucans should not be plural

This was corrected.

187 – please clarify where these “isolates” came from.

Isolate details are provided under “Strain cultivation and conditions” in the Methods section along with the citation of the work in which they were isolated.

190 – this is mentioned elsewhere but if you have GH97s highly represented in your isolates, that could be the source of a1,6 activity. Do you have a citation for the GH13_31 a1,6 activity? Can you purify the enzyme and show this?

We agree that the GH97 could also be a source of a1,6 activity, which is now better represented in the revised Discussion. A reference for the GH31_31 is now provided. We annotated this function based on ~50% structural identity to a characterized GH13_31 from *Bacillus cereus*. As we focused our expression efforts on the GH13 repertoire encoded by our model *Polaribacter* sp. Hel_I_88 and as this organism does not encode a GH13_31, we didn't attempt to express this enzyme.

197 – Glycogen and pullulan contain both a1,4 and a1,6 linkages. Glycogen is not mostly a-1,4-linked. It contains a significant number of a1,6 branch points, more than plant amylopectin.

We have revised this section accordingly. It now reads “alpha-1,4 link-containing glycans glycogen and pullulan”.

199 – Please clarify that this product is likely panose otherwise it seems like this enzyme is a dedicated pullulanase that hydrolyzes a1,6 bonds to produce maltotriose.

This was added to the text for clarity. It now reads “products with a degree of polymerization of three (dp3), likely panose or isopanose”.

216 – until you purify an enzyme with a1,6 activity, I don't think you can say the marine bacteria are adapted towards glucan containing a1,6 bonds.

As the reviewer kindly pointed out, there are multiple enzymes such as GH97 within marine alpha-glucan PULs that may be responsible for the a1,6 activity. All of our annotations of enzymes such as GH97 and GH13_31 point towards this function being exhibited by marine bacteria, and our statement that our results “support an adaptation (...) towards a1,4/a1,6 substrates” was made accordingly.

224 – “linear glycogen” is a bit of a misnomer. The definition of glycogen is that it is a branched molecule. Maybe call it a1,4 glucan at this point.

This was changed to “linear a1,4-glucan”.

231 – “laminarin-targeting Glg-proteins” is a bit confusing. Could you clarify what you meant here?

This was revised for clarity. A missing comma was added.

233 – Is this really a response specific to only laminarin? Would growth on alginate or fucoidan lead to α -glucan synthesis?

We recognize that the focus on laminarin has been somewhat narrow, but nevertheless our analyses indicate that laminarin degradation and glycogen synthesis occur simultaneously within the same bacterial clades, in vitro and during microalgal blooms. We have revised the Discussion to reflect the breadth of algal biomass that could potentially be used, but remain of the view that laminarin is likely to be the major contributor to bacterial α -glucan synthesis. We now elaborate that this is not only because of its importance as the major diatom polysaccharide, but also because it is a glucose homopolymer. Its degradation pathway includes a GH149 that produces Glc-1-phosphate, which is a direct precursor of glycogen biosynthesis and is therefore likely to be energetically favored over synthesis from other monosaccharides, e.g. derived from alginate or fucoidan. We suggest that similar to starch in terrestrial bacteria, the glucose storage β -glucan laminarin, but also α -glucan, are the preferred polysaccharides that are consumed first. We recently observed such a substrate preference for laminarin over alginate and pectin in the marine bacterium *Alteromonas macleodii* (Koch et al. PMID: 30116038). This is not to say that other substrates are not used in this way, but the importance of the glucose-polymer laminarin underlines its role as the primary fuel for this cycle, as demonstrated by our results.

234 – need a “the” before “sole”

This has been corrected.

237 – same question as in line 233

As detailed above, we consider algal laminarin to be one of the most important inputs to the microbial recycling loop during diatom-dominated blooms. The text has been edited to reflect that.

257 – Since this α -glucan storage and release is central to your model, I think it is imperative for you to run some analytical techniques, like NMR, on this purified polysaccharide to determine its structure. Do you simply have bacterial glycogen or is there something special about Flavobacteria intracellular α -glucan?

In this study, we characterized flavobacterial enzymes that are active on different types of α -glucans. These enzymes were similarly active on our enriched intracellular extract, clearly demonstrating that it contains an α -1,4 glucan similar to that on which the enzymes were characterized. These results are also consistent with α -glucan storage polysaccharides described in bacteria in general as well as marine bacteria in particular (PMID: PMID: 26443753; 27768819).

As the reviewer has correctly pointed out, the extract is most likely an enrichment of several bacterial polysaccharides, which is why extensive optimization work would be required before analytical work such as NMR could be performed. As our tested strains were able to grow on the extract and no other gene clusters were induced by growth on the extract apart from the α -glucan PUL (Table S4), it is a viable basis for the model proposed in this study. Of course, the

characterization of the flavobacterial storage polysaccharide and the determination of possible differences from other bacteria is a valuable task for future studies.

277 – I think you meant to cite 5B and not 5A. I would recommend doing qPCR and not proteomics to show this.

As mentioned above, we chose to perform the more time-consuming proteomics experiments in order to gain insight into the actual protein levels produced by the bacteria, and thus make a direct link to observed enzymatic functions.

288 – The difference in α -glucan utilization you observe is likely due to the presence of different GH13s, not the SusDs. If you wish to make the point that SusDs also drive this difference, they need to be expressed, purified and have their α -glucan binding ability characterized.

Thank you for this hint. We agree, the statement now reads “high abundance was not mirrored by the PUL’s associated CAZymes, a GH13 (FG28_RS04360) and a GH65 (FG28_RS04365), and suggests a difference in alpha-glucan utilization”, emphasizing that we consider the CAZymes to be important for the difference in alpha-glucan utilization.

299 – I don’t think it is fair to say that the *Muricauda* strain can “quickly take up and recycle these α -glucans” when it didn’t show very robust growth on extracted α -glucan, pullulan or glycogen.

We agree and have revised the statement to better represent our data. It now reads: “both studied model bacteria can produce, take up and recycle these α -glucans”. As explained above, also *Muricauda* sp. lysate activity on alpha-glucans from cultures grown on glycogen is significantly increased. Furthermore, previous work with this strain indicates that growth on polysaccharide seems to not be preferred, as it also grew significantly better on mannose than different mannans, while still having significant activity on such substrates (PMID: 36411326).

310 – *Muricauda* targeting simple alpha glucans is a bit confusing to me. By simple do you mean shorter or just less complex/branched? Do you think *Muricauda* grows on α -glucan polysaccharide degraded by *Polaribacter*? The *Polaribacter* enzymes kind of go against the “selfish PUL” paradigm if they are capable of degrading α -glucan to DP1/2 and they contain SP2 signal peptides (ie are all on the outside of the cell). Do you think *Muricauda* forages on these DP1/2 products?

This is an interesting suggestion that we think could be a good explanation for the behavior of the *Muricauda* sp. strain and perhaps “open” PUL-type containing strains in general. This is a hypothesis that will be tested in a follow-up study focusing on different niches of alpha-glucan degradation. We have adjusted the text (“which seemingly appears to targeting more simple, e.g. less branched or shorter α -glucans”) to reflect what exactly was meant.

327 – I don’t think it is right to say they are structurally similar when you haven’t characterized the structure of your proposed bacterial α -glucan.

Based on similar results obtained when incubating our analyzed GH13s with extract and alpha-glucans of known structure, we deem it appropriate to state that they are structurally similar.

351 - do you have a reference for this statement?

We have revised this statement to read “Similar to the microalgal β -glucans, abundant bacterial storage alpha-glucans are likely highly soluble in water and as such provide an accessible energy and carbon source”.

376-78 – Is it fair to compare the extracted pigments if they were purified differently? Both methods include a chromatographic separation via HPLC, which is the widely recognized standard (DOI: 10.1016/S0022-0981(97)00141-X). This is done to separate the different pigments to reduce the number of particles in the sample that potentially absorb at similar wavelengths. There are many extraction methods available, and the main property they differ in is the application, especially the type of samples being analyzed. Both our methods are specific to marine environments and were used on the same sampling site.

Furthermore, we use the Chl a data to determine bloom progression to anchor our expression data to and do not compare the values of individual years. Therefore, we deem it appropriate to use the data as a proxy for bloom progression.

412 – stated elsewhere but please update predicted GH13 subfamilies to the current CAZy database.

The database was updated all annotations according to the latest version of the CAZy database as requested.

432 - -1 needs to be in superscript.

This has been corrected.

550 – Please list the source of all oligo and polysaccharides used in your work.

Specifications were added for all oligo- and polysaccharides.

587-589 - Your methods describe cloning of GH13 proteins 1, 2 and 3 but the main text describes them as A, B and C. Please be consistent. Were these cloned without the predicted SP2 signal peptides?

This was an oversight from an older version of the manuscript. It has been remedied. It has been added to the Method section that their signal peptides were omitted for expression.

604 – please also provide final protein concentrations. 25 ug is a lot of protein....especially to let the reaction run for 24 hours.

The reaction was performed for 24 h to reach end-point measurements.

633 – how did you validate that you didn't have any membrane bound polysaccharides?

The text has been modified to reflect that we work with an enriched intracellular polysaccharide fraction, and not a pure polysaccharide. As detailed above, a fraction containing the pellets removed before lysis was analyzed alongside the enriched intracellular extract. This analysis yielded higher amounts of galactosamine, showing that we indeed enriched intracellular polysaccharide with our extraction method.

637 – stated elsewhere but it would be standard practice to characterize your polysaccharide via an initial isoamylase treatment (to remove branch points and allow you to determine the degree of branching and chain length distribution) followed by b-amylase/a-amylase treatment. See PMC7407607

As mentioned above, we do not consider our polysaccharide extract to be pure. As our enzymes were active on the extracted polysaccharide and no proteins other than those of the alpha-glucan PUL were up-regulated during growth on this extract, we did not venture to characterize it further. Of course, work to further purify the extract and determine its exact compositional and structural properties will be part of a follow-up study. However, based on the experiments carried out with GH13A/B/C in this paper, we've gained a first insight into this storage polysaccharide and, importantly, its ecological role.

Minor: the authors provide gene locus tags that don't correlate with protein locus tags. Could the authors provide protein names and what database they are derived from (GenBank, IMG, UniProt?).

Our manuscript contains gene locus tags, not protein identifiers. A reference to the NCBI Refseq assemblies used is now detailed in the Method section.

Reviewer #2

Beidler et al describe an investigation into glucose cycling in the ocean, by means of bacterial α -glucans that appear to be recycled by the bacterioplankton community. With algal laminarin serving as primary input carbon source, bacteria produce α -glucans, undergo cell lysis, and the released α -glucans are taken up by other species, forming a loop. Support is drawn from genomic analyses, growth studies, transcriptomics, proteomics, and enzyme characterisation work. The story is very compelling, and the work represents a major advance, building on previous insightful studies from this cluster of authors looking at the importance of laminarin in the ocean. This work defines an aspect of carbon cycling that had previously been overlooked, and indeed establishes bacterial glycans as a major carbon source for bacterioplankton. While the overall story told in the paper is very compelling, some of the experimental data are not quite convincing in their support of specific points of discussion. Some data points may be over-interpreted when drawing conclusions. And the writing could be improved in places, for better clarity. The points below should be addressed prior to publication, but on the whole this is a very nice and sound piece of work.

We thank the reviewer for the thorough feedback and hope that the changes detailed below improve the overall clarity and scope of the presented manuscript.

Some references are out of date. Reference 10 is a review from 2009, cited in the Introduction as a general discussion on PULs, a topic on which many more reviews have been published in the past 3-5 years. Moreover, the review cited focusses on human gut symbionts, whereas other reviews have been published that also discuss environmental species and may therefore be more relevant. Similarly, reference 57 for the CAZy database is very out of date (2009) and has been superseded more than once. See the CAZy website for guidelines on how to cite the use of the tool today (i.e., there is a paper in Nucleic Acids Res from 2022).

Thank you for pointing this out. We updated the references accordingly.

In the Introduction on pg5, the authors say “In this study, we show that marine Flavobacteriia...degrade the microalgal β -glucan laminarin and use excess glucose to synthesize α -glucan storage polysaccharides.” Where is the evidence in this paper that excess non-metabolised glucose from laminarin goes directly into a pathway for α -glucan synthesis? A similar statement is made in the Discussion on pg13, “Our data suggest that a substantial fraction of this laminarin is not immediately remineralized but rather converted to bacterial α -glucans.” While this is an interesting theory and would fit with the data presented, I don’t agree that it has been proven or even tested in this study. The authors do verify that “laminarin degradation coupled with simultaneous α -glucan-synthesis could be confirmed with an isolated strain in vitro” but I don’t agree that these data show a direct transfer of glucose from metabolised laminarin into an α -glucan synthesis pathway.

We agree that we may have provided that wrong impression that we had direct evidence for the conversion of diatom laminarin to bacterial alpha-glucans, and we rephrased the revised manuscript accordingly. We are now very clear that laminarin-degradation produces the precursor for alpha-glucan synthesis (see also above our statement to reviewer #1 to this point).

Furthermore, we provide a new figure (Fig. 4), in which we show by a combination of deep metagenomics and metatranscriptomics that laminarin-degradation, alpha-glucan synthesis and alpha-glucan degradation occurred simultaneously in members of the flavobacterial genera *Aurantivirga* and *Polaribacter* that dominated the free-living bacterioplankton during the 2020 spring bloom at Helgoland Roads. While not a direct proof, this is a strong indication that the conversion of diatom laminarin to alpha-glucans indeed took place in these bacteria.

The “Helgoland Roads time series” is first mentioned on line 137 and is not defined – please provide some context as to what this is/was.

A more detailed explanation is provided in the revised version of the manuscript.

On line 140 the authors state that “Dinophyceae also known to contain α -glucans...made up 7 to 47% of the eukaryotic population.” Since the abundance of some of these species correlates well with the ‘main phase’ second bloom, why were they not considered another major source of α -glucan substrate, which would disrupt the notion of an almost self-sustaining bacterial loop?

We have revised Figure 1 in order to better showcase the 18S data. We categorized the abundant detected dinoflagellates as “majorly heterotrophic” or “majorly autotrophic” based on the available literature (see Methods and Table S1). The only autotrophic genus that showed up in relevant numbers in our data was *Karenia* and as the text details, this group was most abundant during the early bloom phase (Figure 1C) for which we didn’t detect strong alpha-glucan PUL expression. During the main bloom, autotrophic dinoflagellates played only a minor role (up to 3% of detected non-metazoan reads, Table S1), while heterotrophic dinoflagellates such as *Gyrodinium* showed much higher abundances (up to ~26% of all non-metazoan reads). While these dinoflagellates do most likely contain alpha-glucan as their storage polysaccharide, we detected only a small increase in alpha-glucan on 3 μ m filters, where these dinoflagellates were most abundant (Table S1, Figure S4) while there was a large increase when autotrophic dinoflagellates were present in the early bloom phase. This, together with the fact that heterotrophic dinoflagellates most likely prey on heterotrophic bacteria rather than provide them with nutrients, led us to conclude that dinoflagellate-derived alpha-glucan most likely played only a minor role during the Helgoland bloom of 2020.

Additionally, biovolume data differs from 18S counts as it takes into consideration different cell sizes. So, while some dinoflagellates seem abundant looking only at 18S data, it is important to remember that the respective cell size of many diatoms dwarfs that of most dinoflagellates (see also PMID: 37069671 for previously published biovolume data). As we detail in the discussion, it is likely that dinoflagellate storage polysaccharides can be a C-source for heterotrophic bacteria, but given the massive induction of bacterial alpha-glucan PULs during the main bloom phase, it doesn’t seem likely that the bacteria responded to dinoflagellate alpha-glucan.

In the paragraph beginning line 155, it would be helpful to mention the enzyme specificities typically found for GH13 enzymes, as not all readers will be familiar with the family.

This was added to help understanding for a broad readership.

Lines 163-164, can proteins be classed into functionally distinct groups based only on structure predictions, without any functional characterisation data? The proposed differences in specificity make sense but it appears that the GH13 abundance in PULs with different types of SusD is the only supporting evidence for a difference in binding specificity, since both SusDs seem to have been modelled with the same ligand.

We have added Figure S2, in which we show affinity of the looped-group SusD of *Flavimarina* sp. Hel_I_48 towards alpha-glucans such as glycogen and pullulan but not laminarin. This SusD, which we were able to functionally overexpress and purify, has an almost identical surface structure to the looped SusD of *Polaribacter* sp. Hel_I_88. We have additionally revised Figure 2 to show all binding residues proposed in *B. thetaiotaomicron* to better display our data. The missing loop in the “open”-group contains two binding residues characterized in *B. thetaiotaomicron*, while the other residues are fairly well conserved and therefore gives an indication of a different binding mechanisms between the two SusD groups. Of course, further functional characterization of SusDs belonging to both PUL groups need to be performed as part of a follow-up study regarding the here proposed differences in their binding mechanisms.

Line 172, a small issue of lack of clarity in writing. Is “more GH13s” really an indicator of greater diversity than “some GH13s” ?

The section has been updated to clarify the differences between the two types. It now reads “Gene composition analysis showed that PULs with an open type SusD almost exclusively coded for only few enzymes, e.g. only a sole GH13, whereas PULs with a looped SusD comprised a wider variety of CAZymes, most notably up to four GH13s.”

In the growth studies shown in Figure S1B-C, I would like to see control experiments using glucose and a “no carbon source” unsupplemented medium to see the baseline for minimal possible growth. *Polaribacter* growth on the extract is minimal (and apparently confounded by aggregation issues), and growth for *Muricauda* is at the same minimal level for all experiments shown here. Given the overall low level of growth for *Muricauda*, I don't support the statement on line 181 of the main manuscript that there is evidence of a ‘strong’ preference for glycogen versus ‘poor growth’ on pullulan, since both seem poor. A glucose control experiment would be very helpful for demonstrating ‘good’ growth, and an experiment without any carbon source would clarify whether either species is able to use the extract at all.

Controls with no carbon source or glucose are now provided. Additionally, the text was adapted to better represent the results. Specifically, the “strong” before preference was omitted.

On Figure S1B, consider highlighting (where present) the loop that is absent in the ‘open’ SusD structure.

Figure S1B (now S3) was revised for clarity, highlighting the loop. The figure now also provides AlphaFold confidence scores to display the validity of our model.

The paragraph on line 186-192 is very unclear, please try to re-phrase all of it. I am not sure what was compared to what, or for what purpose.

This was revised to clarify that this analysis included 53 bloom-associated isolates as well as bloom-associated metagenomes.

The three characterised GH13 enzymes are referred to in text as GH13A, GH13B, and GH13C, starting on pg8. Yet the PUL shown in Figure 5 does not indicate which these are in the PUL. And in the Methods section on pg23 (Cloning section) and pg25 (Bacterial glycan characterisation section) they are referred to as GH13 (1), GH13 (2), and GH13 (3). Confusing! Naming consistency is essential.

The GH13s are now referred to as GH13A/B/C throughout the manuscript and are indicated as such in all figures where appropriate.

The FACE results are not always clear, with some showing a “simile/frown” artefact that makes comparison with standards difficult, and many showing a green line across all or most lanes. Since HPAEC-PAD was used for other experiments, why was it not used for oligosaccharide analysis? The data could have been cleaner.

For cyclodextrins and polysaccharides, where the undigested substrate was not visible in the FACE gel analysis, we now provide additional TLC data (new Figure S5), which confirms our previous results. HPAEC-PAD was considered for enzyme characterization but was not performed due to time constraints.

On pg9, line 231, what are the “laminarin-targeting” glg proteins? This paragraph in general is rather unclear and confusing: since Glg proteins are presented as being involved in α -glucan synthesis, whereas “laminarin-targeting” implies degradation. On Figure S3, panel B is labelled with “ α -glucan biosynthesis” and in the legend it is described as showing “Glg-pathway” – how does that relate to laminarin?

Thank you for this hint. We revised this section for clarity. The problem was a missing comma between “laminarin-targeting” and “Glg proteins”. Glg-proteins are involved in alpha-glucan synthesis and clearly do not target laminarin.

The legend of Figure S3 (now S6) was revised to match the figure.

On line 244 it states “...we showed increased amounts of α -glucan present in the culture (Fig S5C).” But Figure S5C shows protein abundance of α -glucan PUL proteins. Where is the data showing the amount of α -glucan accumulating in culture medium?

This has already been discussed in the text (234-237) and refers to the data in Figure S4A (now S7). We have added this distinction in the text to avoid confusion.

More detail is needed about the characterisation of the extract. Figure 4A shows sugar composition of the extract, and the legend implies that this was achieved using enzyme hydrolysis – is that accurate? The Methods section would suggest acid hydrolysis was used, and I assume this was

for analysis of the extract, so the Figure 4 legend should be corrected. In addition, how was the “mol%” content of each sugar determined? Is this representative of the entire sample, or only of the carbohydrate portion of the sample? (i.e. were the amounts of each sugar detected by HPAEC simply added together, or was a % of sample original weight determined?) The purity of the polysaccharide in terms of the absence of protein, lipid, etc is very important in understanding results of the growth study.

The legend of Figure 4 (now 5) was revised for clarity and now clearly states the use of acid hydrolysis. Mol% refers to the percentage of carbohydrate in the sample. This distinction was added to the legend.

As the extract was purified using a chloroform-extraction, where only the aqueous phase but not the organic or proteinaceous interphase was collected. The extract should therefore contain very little protein contamination. Additionally, single strain proteomics showed no upregulation of proteases during growth on the extract (Table S4).

Standards are not labelled on the FACE image shown in Figure 4C.

Standards are now labeled on all figures.

Lines 273-274, where protein abundance is compared to alginate controls. Some of the proteins from the PUL seem quite abundant in the alginate sample, while others do not – how do the authors account for this, and was alginate the most suitable choice as a control here?

The proteins of the alginate control are in many cases (e.g. SusC/D, GH13_38, GH13_10) significantly less abundant than during growth on either extract or glycogen. We have revised Figure 5 (now 6) to better present our data and show the statistical significance of our results. As shown in Figure 6A, lysate from cultures grown on glycogen or extract has a significantly higher activity on alpha-glucans than that of cultures grown on alginate and vice versa. Combined with *Polaribacter* sp. Hel_I_88, which encodes an alginate PUL and grew well on this substrate as the sole carbon source, we considered it an appropriate control condition. In particular, alginate does not contain glucose, minimizing the possibility of cross-reactivity.

Please check your references to Fig 5A and Fig5B – panel A shows enzyme assays for both species, panel B shows protein abundance for both species. (Mis-referenced on line 280 and possibly elsewhere)

The references to this figure in the text were corrected.

Lines 280-282 seem to assume there are no α -glucanases from outside the PUL that are being expressed, have the authors looked at the whole genome to verify this?

As shown in Table S4, *Muricauda* sp. MAR 2010_75 encodes three GH13s that are not encoded within the alpha-glucan PUL (FG28_RS05485, FG28_RS07530, FG28_RS18610). These were checked for differential expression during growth on all substrates tested, but no enzyme showed a significant increase in abundance during growth on glycogen and extract.

The paragraph in lines 299-307 says that the authors “have shown” that α -glucan uptake is “quick” and recycling is “rapid” – where has this been tested in this study? Have the authors for example examined the speed of transcriptional response in the PULs? I don’t see such supporting data.

Thank you for this hint, the section was revised for clarity to reflect our results. We have shown that the bacteria can produce (Fig. S7), take up and recycle alpha-glucans (Table S4, Figure S5). The statement now reads: “both studied model bacteria can produce, take up and recycle these α -glucans”.

Line 319, did the authors consider characterising this presumed α -1,6- specific GH13_31? It would give clear support to the notion of a PUL targeting more complex polysaccharides.

In this study, we focused on characterizing the GH13 repertoire of *Polaribacter* sp. Hel_I_88, as it was a good representation of the majority of the GH13s in isolates and bloom-associated bacterial MAGs. This *Polaribacter* sp. does not encode for a GH13_31, and so we did not venture to characterize this specific enzyme. However, the hits in HMMER and BLAST for the GH13_31 domain are valid and ~50% sequence identity to a characterized enzyme from *Bacillus cereus* (PMID: 9193006) lead us to determine that the enzyme has the proposed function.

In the Methods, I note that the CAZy database was accessed in 2020 for transcript annotation. The database has been updated several times since then, with new families and sub-families being defined very frequently. Is it feasible to re-annotate the data with a more recent database?

Re-annotation of the transcriptomic data was performed as suggested, and all other data were re-annotated with the most recent database released after manuscript submission for consistency.

Line 551-552 – why was a higher concentration of the extract provided compared to other polysaccharides, and does this not skew the growth data?

We cannot rule out that this may have an influence on the growth kinetics. In the first experiments with the extract, 0.1% was used but no growth could be detected. As the enriched polysaccharide extract was not pure (~30% glucose), we then increased the amount added in the growth experiments to account for this impurity, resulting in growth of the culture. Growth experiments were mainly performed to see if a strain could utilize the substrate at all. If the strains could not use the substrate, they would not grow, regardless if 0.1 or 0.2% were used.

Line 569 – I assume (brand) is a holdover from an unfinished version of the manuscript.

This was corrected.

What was the reaction volume used in enzyme characterisation assays (bottom of pg23)? And on line 611-612, what does it mean that assays were compared to solutions with only enzyme OR polysaccharide? Which was used as baseline?

The description of this method has been revised for completeness and clarity. What was meant by this was that all the samples were compared to a sample containing only poly-/oligosaccharide, which was used as a baseline. In addition, samples containing only enzymes were analyzed to see if they increased the signal, which was negative for all enzymes tested.

A GH16 enzyme from *Christiagramia forsetii* is mentioned in a figure legend, but it is not referred to anywhere in the text. There should be explanation somewhere of why this was used.

A citation for this enzyme was added. The distinction that we use this enzyme as a control to check for other glucose-based polysaccharide was added to the text.

Reviewer #3

General comments:

Beidler and colleagues combine field and culture data to assess the role of laminarin degradation products in fueling secondary production in marine bloom events. They use proteomics, carbohydrate analyses, culture experiments and recombinant enzyme characterization to identify the proteins and linkage patterns associated with bacterial usage of algal-derived carbohydrates such as laminarin. The authors are quite thorough in their analyses and pose a compelling set of interpretations from their data. However, my impression is that their overall point is not necessarily supported by the data that they present. The data thoroughly covers the production and consumption of α -glucans by bacteria and their specificity towards different glucan linkages – but it does not cover the field component which is then summarized in Figure 6. Specifically, the temporal evolution of the system is not well-supported by the data presented here. The data is great, but I wonder if it is being used to tell the wrong story, or at least to speculate further than the data can go? For example, in Figure 6 (the summary figure), there are a few places where data seems to be missing and/or incomplete. Is there clear evidence of the temporal separation between steps 3 and 5? My read of the data presented is that the progression from B-glucan PULs to α -glucan PULs is not available here. If that is wrong, perhaps this data could be displayed or described more prominently.

We thank the reviewer for the helpful suggestions to improve the overall scope and clarity of our work and herewith provide additional data as well as improvements to the text and figures that we hope clarifies the levied concerns. The additional transcriptomic data now provided in the new Figure 4 details that laminarin degradation, alpha-glucan synthesis and degradation not only occur simultaneously within the bloom-associated bacteria (as shown in Figure S6 (previously S3)) but that they occur simultaneously within bacteria of the same genera. We have also revised Figure 6 (now 7) to better reflect that the processes discussed in this paper don't seem to occur separated from each other, but rather in a continuous loop.

Data shown in Figure S7 (previously S4A) strongly indicates alpha-glucan synthesis from laminarin, as the culture was grown with only laminarin as sole carbon source and shows a significant alpha-glucan increase over time. Furthermore, we now elaborate that laminarin is not only of importance to this process because it is a major diatom polysaccharide, but also because it is also a glucose homopolymer. Its degradation pathway includes a GH149, that produces Glc-1-phosphate, which is a direct precursor of glycogen/alpha-glucan biosynthesis and is therefore likely to be energetically favored. However, we do realize that the focus on laminarin alone is too narrow and therefore have adjusted the wording in the text as well as in the revised Figure 6 (now 7) to reflect this.

Furthermore, recycling of biomass within the bacterial population has recently been suggested as an overlooked carbon pool using 16s/18s-based in silico modeling, (doi: <https://doi.org/10.1101/2024.01.10.574976>).

Second, it is not clear if the soluble α -glucans depicted in the center of the figure have been observed in seawater by the authors. Certainly glucose and its small polymers are vanishingly dilute in seawater, but do the authors have data on the rates of turnover of this pool? I believe that the

turnover rates could be quite fast, but there is no data to constrain these values and test this hypothesis.

In this work, we have not attempted to observe alpha-glucans in seawater samples. In a previous study, we could detect alpha-glucans in marine POM samples from the western North Atlantic Ocean as well as the North Sea (PMID: 35765187). In this study, we show that the bacterial fractions extracted from filters from the 2020 Helgoland phytoplankton bloom contained high amounts of alpha-glucan (Fig. S9). In a recent study, we could show that the mortality rates of bloom-associated bacteria within a phytoplankton bloom are comparably high to their growth rates, indicating that a significant amount of bacterial biomass is recycled within just one day (PMID: 37195198), as we indicate in the Discussion. This means that the observed alpha-glucans from the bacterial fraction are most likely released as a part of this cycle. Of course, data specific to alpha-glucan turnover should be a part of a future study to determine specific turnover rates. But based on the data cited and the data in this work, we believe that alpha-glucan is a significant contributor to the bacterial necromass being turned over every day.

Finally, the grazing / viral lysis hypothesis for a-glucan production is not tested. There has been some nice work showing that intracellular composition is different between viral-infected and non-infected cells. Can the authors assess whether (or how much) this affects their hypothesis of lysed cells releasing a-glucans into the dissolved pool? Would infected and non-infected cells provide the same approximate amount of a-glucans to the dissolved pool? Does anyone know the answer to this question?

Thank you for this interesting point. While it is true that viral infection affects the intracellular composition of bacterial cells, the observed amount of alpha-glucan in the bacterial fraction on our bloom filters indicates that there is a significant source of alpha-glucan available to be utilized. Our (meta)transcriptomics/proteomics data sets indicate that this happens on a large-scale during phytoplankton blooms.

Related to your comment, recent work has shown an almost constant abundance of bacterial cells at the community level during algal blooms, indicating equal growth and mortality rates (PMID: 37195198). Thus, for every new cell, another cell dies, resulting in a roughly constant number of cells over time. The rate of cell division and death is about 1.9/day. We have not observed any evidence of glucose accumulation in dissolved or particulate organic carbon during algal blooms over three months (PMID: 33608542). In the absence of accumulation of the glucose signal HMWDOM, we can conclude that bacterial synthesis and degradation rates of alpha-glucan are balanced during algal blooms and that these rates are linked to bacterial growth and death rates. We have now added the relevant references to the main text.

In general, then, I am impressed by the work represented in this paper, but I wonder if the authors have stretched the data a little further than is merited. I provide some additional specific comments below to improve the manuscript.

Specific comments:

1. Page 13, paragraph starting Line 331: I am not following the calculations presented here as to the importance of this process for bacterial cycling in marine systems. If these compounds are degradation products of laminarin, then isn't their carbon content accounted for in the laminarin value? My sense of this calculation is that it is based on cellular content prior to release and degradation. I'm not sure that these degradation products constitute an additional pool of carbon. I do think that the rate of degradation is important, particularly if the degradation steps release CO₂ and thus they could constitute an important component of respiration. But the data and calculations presented here do not touch on this topic and so I am not clear on how this calculation helps us consider carbon cycle dynamics in a new way.

As we show, the bacteria most likely utilize the laminarin and produce their own storage alpha-glucan at the same time (as shown in Fig. 4). It is therefore not a degradation product, but a separate, significant pool of carbon that is in itself relevant to the marine carbon cycle as it is available to different organisms than laminarin. We show that we can detect this alpha-glucan when analyzing bloom filters of the bacterial fraction (see Fig. S9).

2. Figure 1 caption: I find the figure captions to be challenging for a non-expert to interpret so I am providing some areas of confusion for myself in the hope that fixing them will make the figures more accessible to a more general audience. For Figures 1C and 1D, why are there no values given for the y-axis? I realize that these are stacked figures, but could there be an example (or representative) y-axis for one of the graphs. Some sense of the range would be helpful for both C and D.

Thank you for this hint. As the reviewer points out, these graphs indeed use a stacked axis. We have updated the axis indicating the values represented and hope that this makes them easier to interpret.

3. Figure 2 caption: The manuscript text referencing this figure discusses GH13 and other enzymes, but then they are not depicted anywhere in the figure. Can this be done? I am not an expert in these figures, so I can't make sense of where GH13 would fall in this tree and thus I do not have a good reference point for the figure.

The Figure 2 as well as the text have been adjusted to better represent the data. Of course, the figure itself only references SusD sequences and the differences in binding residues between the two presented SusD groups. However, the enzymes such as GH13s coded alongside these *susD* genes are highly conserved and correlate well to the type of SusD within the PUL. We therefore infer a difference in alpha-glucan recognition and degradation based on the different SusD-types and their associated conserved sets of enzymes.

4. Figure 5 caption: In A, please describe the different bars in the graph. How does "release equivalent (glucose)" mean in the context of lysate activity (the only descriptor for Figure 5A)? For B, what is the y-axis? (what is riBAQ? This abbreviation is not described in the caption.)

"riBAQ" is now explained in the figure caption for clarity.

A dedicated section for lysate activity measurements was added to the Methods section. In short, the reducing end assay used to measure lysate activity compares the amount of reducing ends

released by a sample containing only polysaccharide to a sample containing polysaccharide and lysate. The measured signal is compared to a standard obtained by performing the assay with known amounts of glucose. So, the lysate can release reducing ends from the polysaccharide it is incubated with on a level equivalent to x mg of glucose. A higher value indicates a higher activity of the lysate on the given polysaccharide.

Reviewer #2 (Remarks to the Author):

The revised version of this manuscript is much improved and I especially appreciate that additional experimental work has been performed. The authors' responses to my comments, and the corresponding changes to the manuscript, are welcome and largely deal with the specific concerns I had raised. I still have a few additional comments, which are detailed below, and which I think need to be addressed before publication, but these are mostly minor.

Overall however, one issue remains with the paper – I still think that there are instances where an idea or speculation is presented as a verified fact. This is an issue that all three reviewers raised with the initial submission, and one that I don't think the authors have deeply reflected on. The new version of the paper is improved in this sense but there are still some areas where the language needs to be toned down to make it clear that an idea is being presented for which evidence still needs to be gathered.

In the final section of the paper, where the Figure 7 model is discussed, I would like to see a reflection from the authors of what further work remains to be done in order to verify those aspects of the model that have not yet been fully tested or verified. This would wrap the paper up in a forward-focused manner, and would leave the reader with confidence as to what has been tested and what remains speculative.

Other specific comments below:

1. One of the reviewers asked you to define the term 'necromass' and I don't think you have quite done that. The text now seems to suggest that necromass = biomass, but these are not synonymous terms. Necromass is specifically dead biomass – please update the relevant text to reflect this definition more accurately.
2. The sub-heading 'Introduction' is missing.
3. The new highlighted sentence on lines 90-92 is an example of text that sounds more confident than it has reason to be – I don't think this exact claim is tested in this study, so please revise the sentence to say that this is 'likely' what happens.
4. Line 119. In the response to reviewers you wrote convincingly about why the Dinophyceae are not considered a source of alpha-glucans in your study – please make that same point here in the actual manuscript.
5. Line 134. Explain why families GH65,31,97 are mentioned, as you explained in the previous line why GH13 is of interest.
6. Line 142. Please amend to "...indicated that these proteins LIKELY bind alpha-glucan."
7. Lines 140-148. SusD proteins do not bind to substrate, they bind to ligand, so it is not sensible to discuss their "substrate binding sites." It should be "ligand binding sites."
8. Line 192 – surely beta-cyclodextrin contains beta-linkages?
9. The new text in lines 262-265 is not clear to me – what was the purpose of doing both acid and enzyme hydrolysis on the same samples, and why do the results indicate alpha-glucans? More explanation is needed here.
10. Line 304-306 – I still believe that this is speculation being presented as fact, and the wording needs to be re-considered. The loop you describe is indeed an interesting part of your model but it does remain untested as of yet. You might say "...bacteria MAY employ a necromass recycling loop that WE SPECULATE IS refueled by algal biomass..."
11. Line 337-338 – what does it mean that biomass 'comes from laminarin'? Do you mean that the biomass is laminarin? Surely 'laminarin comes from biomass' – the sentence needs to be re-phrased.

Reviewer #3 (Remarks to the Author):

Thanks very much for addressing my concerns. The text and figures are much clearer and more accessible to the general reader.

Point-to-point reply to the reviewers' additional comments

Manuscript NCOMMS-23-35735A | Alpha-glucans from bacterial necromass indicate an intra-population loop within the marine carbon cycle | Beidler *et al.*

Reviewer #2 (Remarks to the Author):

The revised version of this manuscript is much improved and I especially appreciate that additional experimental work has been performed. The authors' responses to my comments, and the corresponding changes to the manuscript, are welcome and largely deal with the specific concerns I had raised. I still have a few additional comments, which are detailed below, and which I think need to be addressed before publication, but these are mostly minor.

Overall however, one issue remains with the paper – I still think that there are instances where an idea or speculation is presented as a verified fact. This is an issue that all three reviewers raised with the initial submission, and one that I don't think the authors have deeply reflected on. The new version of the paper is improved in this sense but there are still some areas where the language needs to be toned down to make it clear that an idea is being presented for which evidence still needs to be gathered.

In the final section of the paper, where the Figure 7 model is discussed, I would like to see a reflection from the authors of what further work remains to be done in order to verify those aspects of the model that have not yet been fully tested or verified. This would wrap the paper up in a forward-focused manner, and would leave the reader with confidence as to what has been tested and what remains speculative.

Thank you for taking the time to provide additional feedback. We have added a section to the discussion specifying that future studies could demonstrate the proposed carbon flux and calculate specific turnover rates from simultaneous seawater measurements of different glucose polymers to understand the magnitude of this process. We also express the need for data to address the proposed different niches occupied by bacteria with looped/open type SusD containing alpha-glucan PULs.

Other specific comments below:

1. One of the reviewers asked you to define the term 'necromass' and I don't think you have quite done that. The text now seems to suggest that necromass = biomass, but these are not synonymous terms. Necromass is specifically dead biomass – please update the relevant text to reflect this definition more accurately.

We have further adjusted our definition given in the introduction and now detail that necromass specifically indicates biomass that is no longer part of a living organism.

2. The sub-heading 'Introduction' is missing.

The sub-heading was added.

3. The new highlighted sentence on lines 90-92 is an example of text that sounds more confident than it has reason to be – I don't think this exact claim is tested in this study, so please revise the sentence to say that this is 'likely' what happens.

The distinction "likely" was added to the sentence.

4. Line 119. In the response to reviewers you wrote convincingly about why the Dinophyceae are not considered a source of alpha-glucans in your study – please make that same point here in the actual manuscript.

The section about Dinophyceae in the main text was expanded to include the bacterial response and the previously obtained biovolume data indicating that they are not a significant source of alpha-glucan.

The following text has been added: “As the flavobacteria only greatly responded during the second bloom event where the diatom *Ceratulina pelagica* dominated both 18S rRNA counts and previously collected biovolume data, we surmised that dinoflagellates are likely not a significant source of alpha-glucan to the bacteria.”

5. Line 134. Explain why families GH65,31,97 are mentioned, as you explained in the previous line why GH13 is of interest.

We have added that these genes are usually conserved within alpha-glucan PULs and added an additional reference showing a PUL containing these genes to be upregulated during growth on glucose-polymer substrates.

6. Line 142. Please amend to “...indicated that these proteins LIKELY bind alpha-glucan.” “likely” was added to the sentence.

7. Lines 140-148. SusD proteins do not bind to substrate, they bind to ligand, so it is not sensible to discuss their “substrate binding sites.” It should be “ligand binding sites.”

Thank you for this specific hint. We have changed the corresponding wordings from “substrate” to “ligand”.

8. Line 192 – surely beta-cyclodextrin contains beta-linkages?

As the text details, the alpha-/beta-distinction of cyclodextrins refers to the number of glucose units contained in the ring, not the linkage type. An alpha-cyclodextrin contains six glucose monomers, a beta-cyclodextrin seven. However, they both contain exclusively alpha-1,4-linkages.

9. The new text in lines 262-265 is not clear to me – what was the purpose of doing both acid and enzyme hydrolysis on the same samples, and why do the results indicate alpha-glucans? More explanation is needed here.

This section of the text was revised for clarity and now specifies that we used either enzymatic or acid hydrolysis on the extracts, not both approaches in combination.

10. Line 304-306 – I still believe that this is speculation being presented as fact, and the wording needs to be re-considered. The loop you describe is indeed an interesting part of your model but it does remain untested as of yet. You might say “...bacteria MAY employ a necromass recycling loop that WE SPECULATE IS refueled by algal biomass...”

The text has been adjusted and now reads “It thus seems that during diatom-dominated phytoplankton blooms, bacteria employ a necromass recycling loop that is presumably constantly refueled (...)”

11. Line 337-338 – what does it mean that biomass ‘comes from laminarin’? Do you mean that the biomass is laminarin? Surely ‘laminarin comes from biomass’ – the sentence needs to be re-phrased.

The phrase has been reworded. “(...) suggests that a large proportion of this turned-over biomass is laminarin.”